# Non-delusional Q-learning and Value Iteration

**Tyler Lu**
Google AI
tylerlu@google.com

**Dale Schuurmans**
Google AI
schuurmans@google.com

**Craig Boutilier**
Google AI
cboutilier@google.com

## Abstract

We identify a fundamental source of error in Q-learning and other forms of dynamic programming with function approximation. *Delusional bias* arises when the approximation architecture limits the class of expressible greedy policies. Since standard Q-updates make globally uncoordinated action choices with respect to the expressible policy class, inconsistent or even conflicting Q-value estimates can result, leading to pathological behaviour such as over/under-estimation, instability and even divergence. To solve this problem, we introduce a new notion of *policy consistency* and define a local backup process that ensures global consistency through the use of *information sets*—sets that record constraints on policies consistent with backed-up Q-values. We prove that both the model-based and model-free algorithms using this backup remove delusional bias, yielding the first known algorithms that guarantee optimal results under general conditions. These algorithms furthermore only require polynomially many information sets (from a potentially exponential support). Finally, we suggest other practical heuristics for value-iteration and Q-learning that attempt to reduce delusional bias.

## 1   Introduction

Q-learning is a foundational algorithm in reinforcement learning (RL) [34, 26]. Although Q-learning is guaranteed to converge to an optimal state-action value function (or Q-function) when state-action pairs are explicitly enumerated [34], it is potentially unstable when combined with function approximation (even simple linear approximation) [1, 8, 29, 26]. Numerous modifications of the basic update, restrictions on approximators, and training regimes have been proposed to ensure convergence or improve approximation error [12, 13, 27, 18, 17, 21]. Unfortunately, simple modifications are unlikely to ensure near-optimal performance, since it is NP-complete to determine whether even a linear approximator can achieve small worst-case Bellman error [23]. Developing variants of Q-learning with good worst-case behaviour for standard function approximators has remained elusive.

Despite these challenges, Q-learning remains a workhorse of applied RL. The recent success of *deep* Q-learning, and its role in high-profile achievements [19], seems to obviate concerns about the algorithm's performance: the use of deep neural networks (DNNs), together with various augmentations (such as experience replay, hyperparameter tuning, etc.) can reduce instability and poor approximation. However, deep Q-learning is far from robust, and can rarely be applied successfully by inexperienced users. Modifications to mitigate systematic risks in Q-learning include double Q-learning [30], distributional Q-learning [4], and dueling network architectures [32]. A study of these and other variations reveals surprising results regarding the relative benefits of each under ablation [14]. Still, the full range of risks of approximation in Q-learning has yet to be delineated.

In this paper, we identify a fundamental problem with Q-learning (and other forms of dynamic programming) with function approximation, distinct from those previously discussed in the literature. Specifically, we show that approximate Q-learning suffers from *delusional bias*, in which updates are based on mutually inconsistent values. This inconsistency arises because the Q-update for a state-action pair, $(s, a)$, is based on the *maximum* value estimate over all actions at the next state, which

ignores the fact that the actions so-considered (including the choice of $a$ at $s$) might not be *jointly realizable* given the set of admissible policies derived from the approximator. These "unconstrained" updates induce errors in the target values, and cause a distinct source of value estimation error: Q-learning readily backs up values based on action choices that the greedy policy class cannot realize.

Our first contribution is the identification and precise definition of delusional bias, and a demonstration of its detrimental consequences. From this new perspective, we are able to identify anomalies in the behaviour of Q-learning and value iteration (VI) under function approximation, and provide new explanations for previously puzzling phenomena. We emphasize that delusion is an inherent problem affecting the interaction of Q-updates with constrained policy classes—more expressive approximators, larger training sets and increased computation do not resolve the issue.

Our second contribution is the development of a new *policy-consistent backup* operator that fully resolves the problem of delusion. Our notion of consistency is in the same spirit as, but extends, other recent notions of temporal consistency [5, 22]. This new operator does not simply backup a single future value at each state-action pair, but instead backs up a *set* of candidate values, each with the associated set of policy commitments that justify it. We develop a model-based value iteration algorithm and a model-free Q-learning algorithm using this backup that carefully integrate value- and policy-based reasoning. These methods complement the value-based nature of value iteration and Q-learning with explicit constraints on the policies consistent with generated values, and use the values to select policies from the admissible policy class. We show that in the tabular case with policy constraints—isolating delusion-error from approximation error—the algorithms converge to an optimal policy in the admissible policy class. We also show that the number of information sets is bounded polynomially when the greedy policy class has finite VC-dimension; hence, the algorithms have polynomial-time iteration complexity in the tabular case.

Finally, we suggest several heuristic methods for imposing policy consistency in batch Q-learning for larger problems. Since consistent backups can cause information sets to proliferate, we suggest search heuristics that focus attention on promising information sets, as well as methods that impose (or approximate) policy consistency within batches of training data, in an effort to drive the approximator toward better estimates.

## 2  Preliminaries

A *Markov decision process (MDP)* is defined by a tuple $\mathbf{M} = (S, A, p, p_0, R, \gamma)$ specifying a set of *states* $S$ and *actions* $A$; a transition kernel $p$; an initial state distribution $p_0$; a reward function $R$; and a discount factor $\gamma \in [0, 1]$. A (stationary, deterministic) *policy* $\pi : S \to A$ specifies the agent's action at every state $s$. The state-value function for $\pi$ is given by $V^\pi(s) = \mathbb{E}[\sum_{t \geq 0} \gamma^t R(s_t, \pi(s_t))]$ while the state-action value (or *Q-function*) is $Q^\pi(s, a) = R(s, a) + \gamma \mathbb{E}_{p(s'|s,a)} V^\pi(s')$, where expectations are taken over random transitions and rewards. Given any Q-function, the policy "Greedy" is defined by selecting an action $a$ at state $s$ that maximizes $Q(s, a)$. If $Q = Q^*$, then Greedy is optimal.

When $p$ is unknown, Q-learning can be used to acquire the optimal $Q^*$ by observing trajectories generated by some (sufficiently exploratory) behavior policy. In domains where tabular Q-learning is impractical, *function approximation* is typically used [33, 28, 26]. With function approximation, Q-values are approximated by some function from a class parameterized by $\Theta$ (e.g., the weights of a linear function or neural network). We let $\mathcal{F} = \{f_\theta : S \times A \to \mathbb{R} \mid \theta \in \Theta\}$ denote the set of expressible value function approximators, and denote the class of admissible greedy policies by

$$G(\Theta) = \left\{ \pi_\theta \,\middle|\, \pi_\theta(s) = \operatorname*{argmax}_{a \in A} f_\theta(s, a), \theta \in \Theta \right\}. \tag{1}$$

In such cases, online *Q-learning* at transition $s, a, r, s'$ (action $a$ is taken at state $s$, leading to reward $r$ and next state $s'$) uses the following update given a previously estimated Q-function $Q_\theta \in \mathcal{F}$,

$$\theta \leftarrow \theta + \alpha \Big( r + \gamma \max_{a' \in A} Q_\theta(s', a') - Q_\theta(s, a) \Big) \nabla_\theta Q_\theta(s, a). \tag{2}$$

Batch versions of Q-learning (e.g., fitted Q-iteration, batch experience replay) are similar, but fit a regressor repeatedly to batches of training examples (and are usually more data efficient and stable).

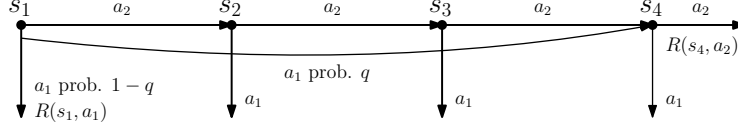

Figure 1: A simple MDP that illustrates delusional bias (see text for details).

## 3 Delusional bias and its consequences

The problem of delusion can be given a precise statement (which is articulated mathematically in Section 4): *delusional bias occurs whenever a backed-up value estimate is derived from action choices that are not realizable in the underlying policy class*. A Q-update backs up values for each state-action pair $(s, a)$ by *independently* choosing actions at the corresponding next states $s'$ via the max operator; this process implicitly assumes that $\max_{a' \in A} Q_\theta(s', a')$ is achievable. However, the update can become inconsistent under function approximation: if no policy in the admissible class can jointly express all past (implicit) action selections, backed-up values do not correspond to Q-values that can be achieved by any expressible policy. (We note that the source of this potential estimation error is quite different than the optimistic bias of maximizing over noisy Q-values addressed by double Q-learning; see Appendix A.5.) Although the consequences of such delusional bias might appear subtle, we demonstrate how delusion can profoundly affect both Q-learning and value iteration. Moreover, these detrimental effects manifest themselves in diverse ways that appear disconnected, but are symptoms of the same underlying cause. To make these points, we provide a series of concrete counter-examples. Although we use linear approximation for clarity, the conclusions apply to any approximator class with finite capacity (e.g., DNNs with fixed architectures), since there will always be a set of $d + 1$ state-action choices that are jointly infeasible given a function approximation architecture with VC-dimension $d < \infty$ [31] (see Theorem 1 for the precise statement).

### 3.1 A concrete demonstration

We begin with a simple illustration. Consider the undiscounted MDP in Fig. 1, where episodes start at $s_1$, and there are two actions: $a_1$ causes termination, except at $s_1$ where it can move to $s_4$ with probability $q$; $a_2$ moves deterministically to the next state in the sequence $s_1$ to $s_4$ (with termination when $a_2$ taken at $s_4$). All rewards are 0 except for $R(s_1, a_1)$ and $R(s_4, a_2)$. For concreteness, let $q = 0.1$, $R(s_1, a_1) = 0.3$ and $R(s_4, a_2) = 2$. Now consider a linear approximator $f_\theta(\phi(s, a))$ with two state-action features: $\phi(s_1, a_1) = \phi(s_4, a_1) = (0, 1)$; $\phi(s_1, a_2) = \phi(s_2, a_2) = (0.8, 0)$; $\phi(s_3, a_2) = \phi(s_4, a_2) = (-1, 0)$; and $\phi(s_2, a_1) = \phi(s_3, a_1) = (0, 0)$. Observe that no $\pi \in G(\Theta)$ can satisfy both $\pi(s_2) = a_2$ and $\pi(s_3) = a_2$, hence the optimal unconstrained policy (take $a_2$ everywhere, with expected value 2) is not realizable. Q-updating can therefore never converge to the unconstrained optimal policy. Instead, the optimal *achievable* policy in $G(\Theta)$ takes $a_1$ at $s_1$ and $a_2$ at $s_4$ (achieving a value of 0.5, realizable with $\theta^* = (-2, 0.5)$).

*Unfortunately, Q-updating is unable to find the optimal admissible policy $\pi_{\theta^*}$ in this example.* How this inability materializes depends on the update regime, so consider online Q-learning (Eq. 2) with data generated using an $\varepsilon$Greedy behavior policy ($\varepsilon = 0.5$). In this case, it is not hard to show that Q-learning must converge to a fixed point $\hat{\theta} = (\hat{\theta}_1, \hat{\theta}_2)$ where $-\hat{\theta}_1 \leq \hat{\theta}_2$, implying that $\pi_{\hat{\theta}}(s_2) \neq a_2$, i.e., $\pi_{\hat{\theta}} \neq \pi_{\theta^*}$ (we also show this for any $\varepsilon \in [0, 1/2]$ when $R(s_1, a_1) = R(s_4, a_2) = 1$; see derivations in Appendix A.1). Instead, Q-learning converges to a fixed point that gives a "compromised" admissible policy which takes $a_1$ at both $s_1$ and $s_4$ (with a value of 0.3; $\hat{\theta} \approx (-0.235, 0.279)$).

This example shows how delusional bias prevents Q-learning from reaching a reasonable fixed-point. Consider the backups at $(s_2, a_2)$ and $(s_3, a_2)$. Suppose $\hat{\theta}$ assigns a "high" value to $(s_3, a_2)$ (i.e., so that $Q_{\hat{\theta}}(s_3, a_2) > Q_{\hat{\theta}}(s_3, a_1)$) as required by $\pi_{\theta^*}$; intuitively, this requires that $\hat{\theta}_1 < 0$, and generates a "high" bootstrapped value for $(s_2, a_2)$. But any update to $\hat{\theta}$ that tries to fit this value (i.e., makes $Q_{\hat{\theta}}(s_2, a_2) > Q_{\hat{\theta}}(s_2, a_1)$) forces $\hat{\theta}_1 > 0$, which is inconsistent with the assumption, $\hat{\theta}_1 < 0$, needed to generate the high bootstrapped value. In other words, any update that moves $(s_2, a_2)$ higher *undercuts the justification* for it to be higher. The result is that the Q-updates compete with each other, with $Q_{\hat{\theta}}(s_2, a_2)$ converging to a compromise value that is not realizable by any policy in $G(\Theta)$.

This induces an inferior policy with lower expected value than $\pi_{\theta*}$. We show in Appendix A.1 that avoiding any backup of these inconsistent edges results in Q-learning converging to the optimal expressible policy. Critically, this outcome is not due to approximation error itself, but the inability of Q-learning to find the value of the optimal *representable* policy.

## 3.2 Consequences of delusion

There are several additional manifestations of delusional bias that cause detrimental outcomes under Q-updating. Concrete examples are provided to illustrate each, but we relegate details to the appendix.

**Divergence:** Delusional bias can cause Q-updating to diverge. We provide a detailed example of divergence in Appendix A.2 using a simple linear approximator. While divergence is typically attributed to the interaction of the approximator with Bellman or Q-backups, the example shows that if we correct for delusional bias, convergent behavior is restored. Lack of convergence due to cyclic behavior (with a lower-bound on learning rates) can also be caused by delusion: see Appendix A.3.

**The Discounting Paradox:** Another phenomenon induced by delusional bias is the *discounting paradox*: given an MDP with a specific discount factor $\gamma_{eval}$, Q-learning with a *different* discount $\gamma_{train}$ results in a Q-function whose greedy policy has better performance, relative to the target $\gamma_{eval}$, than when trained with $\gamma_{eval}$. In Appendix A.4, we provide an example where the paradox is extreme: a policy trained with $\gamma = 1$ is provably *worse* than one trained myopically with $\gamma = 0$, even when evaluated using $\gamma = 1$. We also provide an example where the gap can be made arbitrarily large. These results suggest that treating the discount as hyperparameter might yield systematic training benefits; we demonstrate that this is indeed the case on some benchmark (Atari) tasks in Appendix A.10.

**Approximate Dynamic Programming:** Delusional bias arises not only in Q-learning, but also in approximate dynamic programming (ADP) (e.g., [6, 9]), such as approximate value iteration (VI). With value function approximation, VI performs full state Bellman backups (as opposed to sampled backups as in Q-learning), but, like Q-learning, applies the max operator independently at successor states when computing expected next state values. When these choices fall outside the greedy policy class admitted by the function approximator, delusional bias can arise. Delusion can also occur with other forms of policy constraints (without requiring the value function itself to be approximated).

**Batch Q-learning:** In the example above, we saw that delusional bias can cause convergence to Q-functions that induce poor (greedy) policies in standard online Q-learning. The precise behavior depends on the training regime, but poor behavior can emerge in batch methods as well. For instance, batch Q-learning with experience replay and replay buffer shuffling will induce the same tension between the conflicting updates. Specific (nonrandom) batching schemes can cause even greater degrees of delusion; for example, training in a sequence of batches that run through a batch of transitions at $s_4$, followed by batches at $s_3$, then $s_2$, then $s_1$ will induce a Q-function that deludes itself into estimating the value of $(s_1, a_2)$ to be that of the optimal unconstrained policy.

# 4 Non-delusional Q-learning and dynamic programming

We now develop a provably correct solution that directly tackles the source of the problem: the potential inconsistency of the set of Q-values used to generate a Bellman or Q-backup. Our approach avoids delusion by using *information sets* to track the "dependencies" contained in all Q-values, i.e., the *policy assumptions* required to justify any such Q-value. Backups then prune infeasible values whose information sets are not policy-class consistent. Since backed-up values might be designated inconsistent when new dependencies are added, this *policy-consistent backup* must maintain *alternative* information sets and their corresponding Q-values, allowing the (implicit) backtracking of prior decisions (i.e., max Q-value choices). Such a policy-consistent backup can be viewed as unifying both value- and policy-based RL methods, a perspective we detail in Sec. 4.3.

We develop policy consistent backups in the tabular case while allowing for an arbitrary policy class (or arbitrary policy constraints)—the case of greedy policies with respect to some approximation architecture $f_\theta$ is simply a special case. This allows the method to focus on delusion, without making any assumptions about the specific value approximation. Because delusion is a general phenomenon, we first develop a model-based consistent backup, which gives rise to non-delusional *policy-class value iteration*, and then describe the sample-backup version, *policy-class Q-learning*. Our main theorem establishes the convergence, correctness and optimality of the algorithm (including

---

**Algorithm 1** Policy-Class Value Iteration (PCVI)

---

**Input:** $S, A, p(s' \mid s, a), R, \gamma, \Theta$, initial state $s_0$

1: $\texttt{Q}[sa] \leftarrow$ initialize to mapping $\Theta \mapsto 0$ for all $s, a$
2: $\texttt{ConQ}[sa] \leftarrow$ initialize to mapping $[s \mapsto a] \mapsto 0$ for all $s, a$
3: Update $\texttt{ConQ}[s]$ for all $s$ (i.e., combine all table entries in $\texttt{ConQ}[sa_1], \ldots, \texttt{ConQ}[sa_m]$)
4: **repeat**
5:     **for** all $s, a$ **do**
6:         $\texttt{Q}[sa] \leftarrow R_{sa} + \gamma \bigoplus_{s'} p(s' \mid s, a)\texttt{ConQ}[s']$
7:         $\texttt{ConQ}[sa](Z) \leftarrow \texttt{Q}[sa](X)$ for all $X$ such that $Z = X \cap [s \mapsto a]$ is non-empty
8:         Update $\texttt{ConQ}[s]$ by combining table entries of $\texttt{ConQ}[sa']$ for all $a'$
9:     **end for**
10: **until** $\texttt{Q}$ converges: $\mathrm{dom}(\texttt{Q}(sa))$ and $\texttt{Q}(sa)(X)$ does not change for all $s, a, X$
11: /* Then recover an optimal policy */
12: $X^* \leftarrow \operatorname{argmax}_X \texttt{ConQ}[s_0](X)$
13: $q^* \leftarrow \texttt{ConQ}[s_0](X^*)$
14: $\theta^* \leftarrow \mathsf{Witness}(X^*)$
15: return $\pi_{\theta^*}$ and $q^*$.

---

the complete removal of delusional bias), and computational tractability (subject to a tractable consistency oracle).

## 4.1 Policy-class value iteration

We begin by defining *policy-class value iteration (PCVI)*, a new VI method that operates on collections of information sets to guarantee discovery of the optimal policy in a given class. For concreteness, we specify a policy class using Q-function parameters, which determines the class of realizable greedy policies (just as in classical VI). Proofs and more formal definitions can be found in Appendix A.6. We provide a detailed illustration of the PCVI algorithm in Appendix A.7, walking through the steps of PCVI on the example MDP in Fig. 1.

Assume an MDP with $n$ states $S = \{s_1, \ldots, s_n\}$ and $m$ actions $A = \{a_1, \ldots, a_m\}$. Let $\Theta$ be the parameter class defining Q-functions. Let $\mathcal{F}$ and $G(\Theta)$, as above, denote the class of expressible value functions and admissible greedy policies respectively. (We assume ties are broken in some canonical fashion.) Define $[s \mapsto a] = \{\theta \in \Theta \mid \pi_\theta(s) = a\}$. An *information set* $X \subseteq \Theta$ is a set of parameters (more generally, policy constraints) that justify assigning a particular Q-value $q$ to some $(s, a)$ pair. Below we use the term "information set" to refer both to $X$ and $(X, q)$ as needed.

Information sets will be organized into finite *partitions* of $\Theta$, i.e., a set of non-empty subsets $P = \{X_1, \ldots, X_k\}$ such that $X_1 \cup \cdots \cup X_k = \Theta$ and $X_i \cap X_j = \emptyset$, for all $i \neq j$. We abstractly refer to the elements of $P$ as *cells*. A partition $P'$ is a *refinement* of $P$ if for all $X' \in P'$ there exists an $X \in P$ such that $X' \subseteq X$. Let $\mathcal{P}(\Theta)$ be the set of all finite partitions of $\Theta$. A *partition function* $h : P \to \mathbb{R}$ associates values (e.g., Q-values) with all cells (e.g., information sets). Let $\mathcal{H} = \{h : P \to \mathbb{R} \mid P \in \mathcal{P}(\Theta)\}$ denote the set of all such partition functions. Define the *intersection sum* for $h_1, h_2 \in \mathcal{H}$ to be:

$$(h_1 \oplus h_2)(X_1 \cap X_2) = h_1(X_1) + h_2(X_2), \quad \forall X_1 \in \mathrm{dom}(h_1), X_2 \in \mathrm{dom}(h_2), X_1 \cap X_2 \neq \emptyset.$$

Note that the intersection sum incurs at most a quadratic blowup: $|\mathrm{dom}(h)| \leq |\mathrm{dom}(h_1)| \cdot |\mathrm{dom}(h_2)|$.

The methods below require an *oracle* to check whether a policy $\pi_\theta$ is consistent with a set of state-to-action constraints: i.e., given $\{(s, a)\} \subseteq S \times A$, whether there exists a $\theta \in \Theta$ such that $\pi_\theta(s) = a$ for all pairs. We assume access to such an oracle, "Witness". For *linear* Q-function parameterizations, Witness can be implemented in polynomial time by checking the consistency of a system of linear inequalities.

PCVI, shown in Alg. 1, computes the optimal policy $\pi_{\theta^*} \in G(\Theta)$ by using information sets and their associated Q-values organized into partitions (i.e., partition functions over $\Theta$). We represent Q-functions using a table $\texttt{Q}$ with one entry $\texttt{Q}[sa]$ for each $(s, a)$ pair. Each such $\texttt{Q}[sa]$ is a partition function over $\mathrm{dom}(\texttt{Q}[sa]) \in \mathcal{P}(\Theta)$. For each $X_i \in \mathrm{dom}(\texttt{Q}[sa])$ (i.e., for each information set $X_i \subseteq \Theta$ associated with $(s, a)$), we assign a unique Q-value $\texttt{Q}[sa](X_i)$. Intuitively, the Q-value $\texttt{Q}[sa](X_i)$ is

*justified* only if we limit attention to policies $\{\pi_\theta : \theta \in X_i\}$. Since $\mathsf{dom}(\mathbb{Q}[sa])$ is a partition, we have a Q-value for *any realizable policy*. (The partitions $\mathsf{dom}(\mathbb{Q}[sa])$ for each $(s, a)$ generally differ.)

$\mathtt{ConQ}[sa]$ is a restriction of $\mathbb{Q}[sa]$ obtained by intersecting each cell in its domain, $\mathsf{dom}(\mathbb{Q}[sa])$, with $[s \mapsto a]$. In other words, $\mathtt{ConQ}[sa]$ is a partition function defined on some partition of $[s \mapsto a]$ (rather than all of $\Theta$), and represents Q-values of cells that are consistent with $[s \mapsto a]$. Thus, if $X_i \cap [s \mapsto a] = \emptyset$ for some $X_i \in \mathsf{dom}(\mathbb{Q}[sa])$, the corresponding Q-value disappears in $\mathtt{ConQ}[sa]$. Finally, $\mathtt{ConQ}[s] = \cup_a \mathtt{ConQ}[sa]$ is the partition function over $\Theta$ obtained by collecting all the "restricted" action value functions. Since $\cup_a [s \mapsto a]$ is a partition of $\Theta$, so is $\mathtt{ConQ}[s]$.

The key update in Alg. 1 is Line 6, which jointly updates all $Q$-values of the relevant sets of policies in $G(\Theta)$. Notice that the maximization typically found in VI is *not present*—this is because the operation computes and *records* Q-values for *all choices of actions at the successor state $s'$*. This is the key to allowing VI to maintain consistency: if a future Bellman backup is inconsistent with some previous max-choice at a reachable state, the corresponding cell will be pruned and an alternative maximum will take its place. Pruning of cells, using the oracle Witness, is implicit in Line 6 (pruning of $\oplus$) and Line 7 (where non-emptiness is tested).[1] Convergence of PCVI requires that each $\mathbb{Q}[sa]$ table—both its partition and associated Q-value—converge to a fixed point.

**Theorem 1** (PCVI Theorem). *PCVI (Alg. 1) has the following guarantees:*

> (a) *(Convergence and correctness) The function $\mathbb{Q}$ converges and, for each $s \in S, a \in A$, and any $\theta \in \Theta$: there is a unique $X \in \mathsf{dom}(\mathbb{Q}[sa])$ such that $\theta \in X$ and $Q^{\pi_\theta}(s, a) = \mathbb{Q}[sa](X)$.*
>
> (b) *(Optimality and non-delusion) $\pi_{\theta^*}$ is an optimal policy within $G(\Theta)$ and $q^*$ is its value.*
>
> (c) *(Runtime bound) Assume $\oplus$ and non-emptiness checks (lines 6 and 7) have access to Witness. Let $\mathcal{G} = \{g_\theta(s, a, a') := \mathbf{1}[f_\theta(s, a) - f_\theta(s, a') > 0], \forall s, a \neq a' \mid \theta \in \Theta\}$. Each iteration of Alg. 1 runs in time $O(nm \cdot [\binom{m}{2} n]^{2 \, \mathsf{VCDim}(\mathcal{G})} (m - 1)w)$ where $\mathsf{VCDim}(\cdot)$ denotes the VC-dimension of a set of boolean-valued functions, and $w$ is the worst-case running time of Witness (with at most $nm$ state-action constraints). Combined with Part (a), if $\mathsf{VCDim}(\mathcal{G})$ is finite then $\mathbb{Q}$ converges in time polynomial in $n, m$ and $w$.*

**Corollary 2.** *Alg. 1 runs in polynomial time for linear greedy policies. It runs in polynomial time in the presence of a polynomial time Witness for deep Q-network (DQN) greedy policies.*

(A more complete statement of the Cor. 2 is found in Appendix A.6.) The number of cells in a partition may be significantly less than suggested by the bounds, as it depends on the reachability structure of the MDP. For example, in an MDP with only self-transitions, the partition for each state has a single cell. We note that Witness is tractable for linear approximators, but is NP-hard for DNNs [7]. The poly-time result in Cor. 2 does not contradict the NP-hardness of finding a linear approximator with small worst-case Bellman error [23], since nothing is asserted about the Bellman error and we are treating the approximator's VC-dimension as a constant.

**Demonstrating PCVI:** We illustrate PCVI with a simple example that shows how poorly classical approaches can perform with function approximation, even in "easy" environments. Consider a simple deterministic grid world with the 4 standard actions and rewards of $0$, except $1$ at the top-right, $2$ at the bottom-left, and $10$ at the bottom-right corners; the discount is $\gamma = 0.95$. The agent starts at the top-left. The optimal policy is to move down the left side to the left-bottom corner, then along the bottom to the right bottom corner, then staying. To illustrate the effects of function approximation, we considered linear approximators defined over *random* feature representations: feature vectors were produced for each state-action pair by drawing independent standard normal values.

Fig. 2 shows the *estimated* maximum value achievable from the start state produced by each method (dark lines), along with the actual expected value achieved by the greedy policies produced by each method (light lines). The left figure shows results for a $4 \times 4$ grid with 4 random features, and the right for a $5 \times 5$ grid with 5 random features. Results are averaged over 10 runs with different random feature sets (shared by the algorithms). Surprisingly, even when the linear approximator can support near-optimal policies, classical methods can utterly fail to realize this possibility: in 9 of 10 trials ($4 \times 4$) and 10 of 10 trials ($5 \times 5$) the classical methods produce greedy policies with an expected value of *zero*, while PCVI produces policies with value *comparable to the global optimum*.

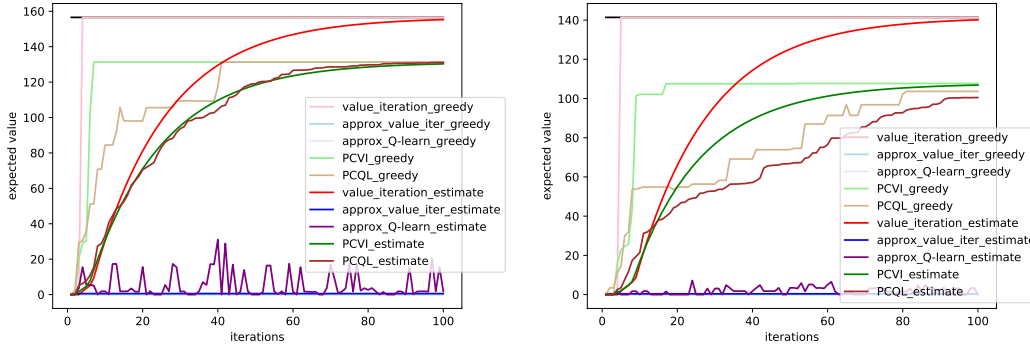

Figure 2: Planning and learning in a grid world with random feature representations. (Left: $4 \times 4$ grid using 4 features; Right: $5 \times 5$ grid using 5 features.) Here "iterations" means a full sweep over state-action pairs, except for Q-learning and PCQL, where an iteration is an episode of length $3/(1-\gamma) = 60$ using $\varepsilon$Greedy exploration with $\varepsilon = 0.7$. Dark lines: estimated maximum achievable expected value. Light lines: actual expected value achieved by greedy policy.

## 4.2 Policy-class Q-learning

A tabular version of Q-learning using the same partition-function representation of Q-values as in PCVI yields *policy-class Q-learning PCQL*, shown in Alg. 2.[2] The key difference with PCVI is simply that we use *sample* backups in Line 4 instead of full Bellman backups as in PCVI.

---

**Algorithm 2** Policy-Class Q-learning (PCQL)

---

**Input:** Batch $B = \{(s_t, a_t, r_t, s'_t)\}_{t=1}^T$, $\gamma$, $\Theta$, scalars $\alpha_t^{sa}$.
1: **for** $(s, a, r, s') \in B$, $t$ is iteration counter **do**
2:    For all $a'$, if $s'a' \notin \texttt{ConQ}$ then initialize $\texttt{ConQ}[s'a'] \leftarrow ([s' \mapsto a'] \mapsto 0)$.
3:    Update $\texttt{ConQ}[s']$ by combining $\texttt{ConQ}[s'a'](X)$, for all $a', X \in \text{dom}(\texttt{ConQ}[s'a'])$
4:    $\texttt{Q}[sa] \leftarrow (1 - \alpha_t^{sa})\texttt{Q}[sa] \oplus \alpha_t^{sa}(r + \gamma\texttt{ConQ}[s'])$
5:    $\texttt{ConQ}[sa](Z) \leftarrow \texttt{Q}[sa](X)$ for all $X$ such that $Z = X \cap [s \mapsto a]$ is non-empty
6: **end for**
7: Return $\texttt{ConQ}$, $\texttt{Q}$

---

The method converges under the usual assumptions for Q-learning: a straightforward extension of the proof for PCVI, replacing full VI backups with Q-learning-style sample backups, yields the following:

**Theorem 3.** *The (a) convergence and correctness properties and (b) optimality and non-delusion properties associated with the PCVI Theorem 1 hold for PCQL, assuming the usual sampling requirements, the Robbins-Monro stochastic convergence conditions on learning rates $\alpha_t^{sa}$ and access to the* Witness *oracle.*

**Demonstrating PCQL:**   We illustrate PCQL in the same grid world tasks as before, again using random features. Figure 2 shows that PCQL achieves comparable performance to PCVI, but with lighter time and space requirements, and is still significantly better than classical methods.

We also applied PCQL to the initial illustrative example in Fig. 1 with $R(s_1, a_1) = 0.3$, $R(s_4, a_2) = 2$ and uniform random exploration as the behaviour policy, adding the use of a value approximator (a linear regressor). We use a heuristic that maintains a global partition of $\Theta$ with each cell $X$ holding a regressor $\texttt{ConQ}(s, a; w_X)$, for $w_X \in \Theta$ predicting the consistent Q-value at $s, a$ (see details in Sec. 5 and Appendix A.8). The method converges with eight cells corresponding to the realizable policies. The policy (equivalence class) is $\texttt{ConQ}(s_1, \pi_X(s_1); w_X)$ where $\pi_X(s_1)$ is the cell's action

at $s_1$; the value is $w_X \cdot \phi(s, \pi_X(s_1))$. The cell $X^*$ with the largest such value at $s_1$ is indeed the optimal realizable policy: it takes $a_1$ at $s_1$ and $s_2$, and $a_2$ elsewhere. The regressor $w_{X^*} \approx (-2, 0.5)$ fits the consistent Q-values perfectly, yielding optimal (policy-consistent) Q-values, because `ConQ` need not make tradeoffs to fit inconsistent values.

### 4.3   Unification of value- and policy-based RL

We can in some sense interpret PCQL (and, from the perspective of model-based approximate dynamic programming, PCVI) as unifying value- and policy-based RL. One prevalent view of value-based RL methods with function approximation, such Q-learning, is to find an approximate value function or Q-function (VF/QF) with low *Bellman error* (BE), i.e., where the (pointwise) difference between the approximate VF/QF and its Bellman backup is small. In approximate dynamic programming one often tries to minimize this directly, while in Q-learning, one usually fits a regression model to minimize the mean-squared temporal difference (a sampled form of Bellman error minimization) over a training data set. One reason for this emphasis on small BE is that the max norm of BE can be used to directly bound the (max norm) loss of the value of *greedy policy* induced by the approximate VF/QF and the value of the *optimal policy*. It is this difference in performance that is of primary interest.

Unfortunately, the bounds on induced policy quality using the BE approximation are quite loose, typically $2||\mathrm{BE}||_\infty/(1-\gamma)$ (see [6], bounds with $\ell_p$ norm are similar [20]). As such, minimizing BE does not generally provide policy guarantees of practical import (see, e.g., [11]). As we see in the cases above (and also in the appendix) that involve delusional bias, a small BE can in fact be rather misleading with respect to the induced policy quality. For example, Q-learning, using least squares to minimize the TD-error as a proxy for BE, often produces policies of poor quality.

PCQL and PCVI take a different perspective, embracing the fact that the VF/QF approximator strictly limits that class of greedy policies that can be realized. In these algorithms, no Bellman backup or Q-update ever involves values that cannot be realized by an admissible policy. This will often result in VFs/QFs with greater BE than their classical counterparts. But, in the exact tabular case, we derive the true value of the induced (approximator-constrained) policy and guarantee that it is optimal. In the regression case (see Sec. 5), we might view this as attempting to minimize BE *within the class of admissible policies*, since we only regress toward policy-consistent values.

The use of information sets and consistent cells effectively means that PCQL and PCVI are engaging in policy search—indeed, in the algorithms presented here, they can be viewed as enumerating all consistent policies (in the case of Q-learning, distinguishing only those that might differ on sampled data). In contrast to other policy-search methods (e.g., policy gradient), both PCQL and PCVI use (sampled or full) Bellman backups to direct the search through policy space, while simultaneously using policy constraints to limit the Bellman backups that are actually realized. They also use these values to select an optimal policy from the feasible policies generated within each cell.

## 5   Toward practical non-delusional Q-learning

The PCVI and PCQL algorithms can be viewed as constructs that demonstrate how delusion arises and how it can be eliminated in Q-learning and approximate dynamic programming by preventing inadmissible policy choices from influencing Q-values. However, the algorithms maintain information sets and partition functions, which is impractical with massive state and action sets. In this section, we suggest several heuristic methods that allow the propagation of some dependency information in practical Q-learning to mitigate the effects of delusional bias.

**Multiple regressors:**  With multiple information sets (or cells), we no longer have a unique set of labels with which to fit an approximate Q-function regressor (e.g., DNN or linear approximator). Instead, each cell has its own set of labels. Thus, if we maintain a global collection of cells, each with its own Q-regressor, we have a set of approximate Q-functions that give both a compact representation and the ability to generalize across state-action pairs for any set of policy consistent assumptions. This works in both batch and pure online Q-learning (see Appendix A.8 for details.)

The main challenge is the proliferation of information sets. One obvious way to address this is to simply limit the total number of cells and regressors: given the current set of regressors, at any update, we first create the (larger number of) new cells needed for the new examples, fit the regressor for

each new *consistent* cell, then prune cells according to some criterion to keep the total number of regressors manageable. This is effectively a search through the space of information sets and can be managed using a variety of methods (branch-and-bound, beam search, etc.). Criteria for generating, sampling and/or pruning cells can involve: (a) the magnitude to the Q-labels (higher expected values are better); (b) the constraints imposed by the cell (less restrictive is better, since it minimizes future inconsistency); the diversity of the cell assignments (since the search frontier is used to manage "backtracking").

If cell search maintains a restricted frontier, our cells may no longer cover all of policy space (i.e, Q is no longer a partition of $\Theta$). This runs the risk that some future Q-updates may not be consistent with *any* cell. If we simply ignore such updates, the approach is *hyper-vigilant*, guaranteeing policy-class consistency at the expense of losing training data. An alternative relaxed approach is to merge cells to maintain a full partition of policy space (or prune cells and in some other fashion relax the constraints of the remaining cells to recover a partition). This *relaxed* approach ensures that all training data is used, but risks allowing some delusion to creep into values by not strictly enforcing all Q-value dependencies.

**Q-learning with locally consistent data:** An alternative approach is to simply maintain a single regressor, but ensure that any batch of Q-labels is self-consistent before updating the regressor. Specifically, given a batch of training data and the current regressor, we first create a single set of consistent labels for each example (see below), then update the regressor using these labels. With no information sets, the dependencies that justified the previous regressor are not accounted for when constructing the new labels. This may allow delusion to creep in; but the aim is that this heuristic approach may mitigate its effects since each new regressor is at least "locally" consistent with respect to its own updates. Ideally, this will keep the sequence of approximations in a region of $\theta$-space where delusional bias is less severe. Apart from the use of a consistent labeling procedure, this approach incurs no extra overhead relative to Q-learning.

**Oracles and consistent labeling:** The first approach above requires an oracle, Witness, to test consistency of policy choices, which is tractable for linear approximators (linear feasibility test), but requires solving an integer-quadratic program when using DQN (e.g., a ReLU network). The second approach needs some means for generating consistent labels. Given a batch of examples $B = \{(s_t, a_t, r_t, s_t')\}_{t=1}^T$, and a current regressor $\widetilde{Q}$, labels are generated by selecting an $a_t'$ for each $s_t'$ as the max. The selection should satisfy: (a) $\cap_t[s_t' \mapsto a_t'] \neq \emptyset$ (i.e., selected max actions are mutually consistent); and (b) $[s_t \mapsto a_t] \cap [s_t' \mapsto a_t'] \neq \emptyset$, for all $t$ (i.e., choice at $s_t'$ is consistent with taking $a_t$ at $s_t$). We can find a consistent labeling maximizing some objective (e.g., sum of resulting labels), subject to these constraints. For a linear approximator, the problem can be formulated as a (linear) mixed integer program (MIP); and is amenable to several heuristics (see Appendix A.9).

## 6   Conclusion

We have identified *delusional bias*, a fundamental problem in Q-learning and approximate dynamic programming with function approximation or other policy constraints. Delusion manifests itself in different ways that lead to poor approximation quality or divergence for reasons quite independent of approximation error itself. Delusional bias thus becomes an important entry in the catalog of risks that emerge in the deployment of Q-learning. We have developed and analyzed a new policy-class consistent backup operator, and the corresponding model-based PCVI and model-free PCQL algorithms, that fully remove delusional bias. We also suggested several practical heuristics for large-scale RL problems to mitigate the effect of delusional bias.

A number of important direction remain. The further development and testing of practical heuristics for policy-class consistent updates, as well as large-scale experiments on well-known benchmarks, is critical. This is also important for identifying the prevalence of delusional bias in practice. Further development of practical consistency oracles for DNNs and consistent label generation is also of interest. We are also engaged in a more systematic study of the discounting paradox and the use of the discount factor as a hyper-parameter.

## Footnotes

[1]If arbitrary policy constraints are allowed, there may be no feasible policies, in which case Witness will prune each cell immediately, leaving no Q-functions, as desired.

[2]PCQL uses the same type of initialization and optimal policy extraction as PCVI; details are omitted.

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
