[Supplementary Material]

# A  Appendix: supplementary material

## A.1  Example 1: Q-learning fixed point derivation

We first characterize the set of fixed points, $\hat{\theta} = (\hat{\theta}_1, \hat{\theta}_2)$, produced by online Q-learning (or QL). (Note that the initial parameters $\theta^{(0)}$ do not influence this analysis.) Using the $\varepsilon$-greedy behaviour policy $\pi_b = \varepsilon\text{Greedy}$, any fixed point must satisfy two conditions:
**(1)** The behaviour policy must not change with each update; If $\pi_{\hat{\theta}}(s) = a$ then $\varepsilon\text{Greedy}$ takes $a$ with probability $1 - \varepsilon$ and any other action $a' \neq a$ with uniform probability.
**(2)** The expected update values for $\theta$ summed across all state-action pairs must be zero under the stationary visitation frequencies $\mu(s, a)$ of the behaviour policy.

The second condition imposes the following set of constraints,

$$\sum_{s,a} \mu(s,a)\phi(s,a)\Big(R(s,a) + \gamma \sum_{s'} p(s'|s,a) \max_{a'} Q_{\hat{\theta}}(s',a') - Q_{\hat{\theta}}(s,a)\Big) = \mathbf{0} . \tag{3}$$

In Fig. 1, we first check if there exist a fixed point that corresponds to the optimal policy. The optimal policy takes $a_1$ in $s_1$ and $a_2$ in $s_4$ deriving expected value $R(s_1, a_1) + qR(s_4, a_2)$ if $s_1$ is the initial state. For linear function approximators, this implies $\theta_2 \geq 0.8\theta_1$ (choose $a_1$ over $a_2$ in $s_1$) and $-\theta_1 > \theta_2$ (choose $a_2$ over $a_1$ in $s_4$), which then implies $\theta_1 < 0$ (hence the policy takes $a_1$ in $s_2$ and $a_2$ in $s_3$). Under $\varepsilon\text{Greedy}$, the stationary visitation frequencies $\mu$ is given in Table 1.

Table 1: $\mu(s,a)$ is the expected number of times action $a$ is taken at state $s$ in an episode under $\varepsilon\text{Greedy}$ for the optimal (greedy) policy.

| $s,a$ | $\mu(s,a)$ | $s,a$ | $\mu(s,a)$ |
|---|---|---|---|
| $s_1, a_1$ | $1 - \varepsilon$ | $s_3, a_1$ | $\varepsilon^3$ |
| $s_1, a_2$ | $\varepsilon$ | $s_3, a_2$ | $\varepsilon^2(1-\varepsilon)$ |
| $s_2, a_1$ | $\varepsilon(1-\varepsilon)$ | $s_4, a_1$ | $\varepsilon^3(1-\varepsilon) + \varepsilon(1-\varepsilon)q$ |
| $s_2, a_2$ | $\varepsilon^2$ | $s_4, a_2$ | $\varepsilon^2(1-\varepsilon)^2 + (1-\varepsilon)^2 q$ |

We can calculate the expected update at each state-action pair, i.e.

$$\Delta\theta(s,a) = \mu(s,a)\phi(s,a)\Big(R(s,a) + \gamma \sum_{s'} p(s'|s,a) \max_{a'} Q_{\theta}(s',a') - Q_{\theta}(s,a)\Big) .$$

For example, at $(s_1, a_1)$ we can transition to either a terminal state (with probability $1 - q$) or $s_4$ (with probability $q$ and after which the best action is $a' = a_2$ with bootstrapped value $-\theta_1$). Therefore at steady state the expected update at $(s_1, a_1)$ is

$$[(1 - \varepsilon)(1 - q)(R(s_1, a_1) - \theta_2) + (1 - \varepsilon)q(R(s_1, a_1) - \theta_1 - \theta_2)] \, \phi(s_1, a_1) .$$

Table 2 lists the expected updates for each state-action pair. If we sum the expected updates and set to $(0,0)$ we get two equations with two unknowns, from which we solve for $\theta$. The equation from the first component implies

$$\theta_1 = \frac{-(1-\varepsilon)^2(\varepsilon^2 + q)R(s_4, a_2)}{(1-\varepsilon)^2(\varepsilon^2 + q) + 0.64\varepsilon + 1.44\varepsilon^2} .$$

The equation from the second component implies

$$\theta_2 = \frac{(1-q)R(s_1, a_1) + qR(s_1, a_1) - q\theta_1}{1 + \varepsilon(\varepsilon^2 + q)} .$$

If we let $R(s_1, a_1) = 0.3$, $R(s_4, a_2) = 2$ and $\varepsilon = 1/2$ then $\hat{\theta} \approx (-0.114, 0.861)$, contradicting the constraint that $\theta_2 < -\theta_1$ ($a_2$ is taken at $s_4$). Fig. 3 shows that for $R(s_1, a_1) = R(s_4, a_2) = 1$, $\theta_2 \not< -\theta_1$ for all $\varepsilon \in [0, 1/2]$ and again violating the assumed constraints. Therefore we conclude QL does not converge to the optimal greedy policy for these parameter configurations.

Now consider if QL may converge to the second best policy where $\pi_\theta(s_1) = a_1$ and $\pi_\theta(s_4) = a_1$ deriving expected value of $R(s_1, a_1)$ starting at state $s_1$. Say $\theta_1 < 0$. The corresponding state-action visitation frequencies at convergence is identical to Table 1 except $\mu(s_4, a_1) = (\varepsilon^2 + q)(1 - \varepsilon)^2$ and $\mu(s_4, a_2) = \varepsilon(1 - \varepsilon)(\varepsilon^2 + q)$. The expected updates are identical to Table 2 except at $(s_1, a_1)$ it is $(0, (1 - \varepsilon)(1 - q)(R(s_1, a_1) - \theta_2) + (1 - \varepsilon)qR(s_1, a_1))$, at $(s_3, a_2)$ it is $(-\varepsilon^2(1 - \varepsilon)(\theta_1 + \theta_2), 0)$, at $(s_4, a_1)$ it is $(0, -(1 - \varepsilon)^2(\varepsilon^2 + q)\theta_2)$ and at $(s_4, a_2)$ it is $(-\varepsilon(1 - \varepsilon)(\varepsilon^2 + q)(R(s_4, a_2) + \theta_1), 0)$. Again, we can solve these equations and get

$$\theta_2 = \frac{-R(s_1, a_1)}{\varepsilon q + \varepsilon^3 - \varepsilon^2 - 1},$$

$$\theta_1 = \frac{-\varepsilon^2(1 - \varepsilon)\theta_2 - (1 - \varepsilon)^2(\varepsilon^2 + q)R(s_4, a_2)}{-(1 - \varepsilon)^2(\varepsilon^2 + q) - \varepsilon^2(1 - \varepsilon) - 1.44\varepsilon^2 - 0.64\varepsilon}.$$

Plugging in $R(s_1, a_1) = 0.3$, $R(s_4, a_2) = 2$ and $\varepsilon = 1/2$ we have $\hat\theta \approx (-0.235, 0.279)$ which is a feasible solution. In particular, we empirically verified that starting with initial $\theta^{(0)} = (0, 0)$, QL converges to this solution. This second best policy is also a fixed point for $R(s_1, a_1) = R(s_4, a_2) = 1$ with any $\varepsilon \in [0, 1/2]$.

The delusion is caused by the backup at $(s_2, a_2)$. Since we assume $\theta_1 < 0$ the bootstrapped future Q-value is $Q_\theta(s_3, a_2)$. But this is inconsistent: there is no $\theta$ taking $a_2$ at $s_2$ and $a_2$ at $s_3$. Such a backup increases $\theta_2$ in order to propagate a higher value along two edges that are inconsistent (these edges can never belong to the same policy, and can pollute the value of the Q-learned policy). In fact, if we enforce consistency by backing up $(s_3, a_1)$, the expected update at $(s_2, a_2)$ reduces from $-1.44\varepsilon^2\theta_1$ to $-0.64\varepsilon^2\theta_1$. The consistent backup reduces two opposing effects: backup at $(s_2, a_2)$ wants to increase $\theta_1$ while backup at $(s_3, a_2)$ wants to decrease $\theta_1$—resulting in a compromise that corresponds to an inferior policy. When $R(s_1, a_1) = 0.3$, $R(s_4, a_2) = 2$, and $\varepsilon = 1/2$ this reduction in the expected update quantity implies QL (with a consistent backup) converges to the optimal policy with $\hat\theta \approx (-0.308, 0.282)$; whereas with an inconsistent backup it converges to the second best policy (and cannot converge to the optimal policy for any initial condition).

Table 2: Expected update $\Delta\theta(s, a)$ for each state-action pair under the optimal greedy policy.

| $s, a$ | Expected update $\Delta\theta(s, a)$ |
|---|---|
| $s_1, a_1$ | $(0, (1 - \varepsilon)(1 - q)(R(s_1, a_1) - \theta_2) + (1 - \varepsilon)q(R(s_1, a_1) - \theta_1 - \theta_2))$ |
| $s_1, a_2$ | $(-0.64\varepsilon\theta_1, 0)$ |
| $s_2, a_1$ | $(0, 0)$ |
| $s_2, a_2$ | $(-1.44\varepsilon^2\theta_1, 0)$ |
| $s_3, a_1$ | $(0, 0)$ |
| $s_3, a_2$ | $(0, 0)$ |
| $s_4, a_1$ | $(0, -\varepsilon(1 - \varepsilon)(\varepsilon^2 + q)\theta_2)$ |
| $s_4, a_2$ | $(-(1 - \varepsilon)^2(\varepsilon^2 + q)(R(s_4, a_2) + \theta_1), 0)$ |

Figure 3: This shows $\theta_1 + \theta_2 > 0$, implying QL cannot converge to the optimal greedy policy for various $\varepsilon$ exploration probabilities. Here, $R(s_1, a_1) = R(s_4, a_2) = 1$ and $q = 0.1$.

## A.2 Divergence due to delusional bias

We show that delusional bias can actually lead to *divergence* of Q-learning with function approximation. In fact, a trivial example suffices. Consider a deterministic MDP with states $S = \{s_1, s_2\}$ and actions $A = \{a_1, a_2\}$, where from any state, action $a_1$ always transitions to $s_1$ and action $a_2$ always transitions to state $s_2$. Rewards are always 0.

Define a linear approximator for the Q-function by a single basis feature $\phi$; i.e. we approximate $Q(s, a)$ by $\phi(s, a)\theta$ for some scalar $\theta$. Let $Z = \sqrt{12 + 2\eta + \eta^2}$ for $\eta > 0$ arbitrarily close to $0$ ($\eta$ is used merely for tie breaking). Then define $\phi(s_1, a_1) = (1 + \eta)/Z$; $\phi(s_1, a_2) = 1/Z$; $\phi(s_2, a_1) = 1/Z$; and $\phi(s_2, a_2) = 3/Z$, which ensures $\|\phi\|_2 = 1$. Clearly, this Q-approximation severely restricts the set of expressible greedy policies: for $\theta > 0$ the greedy policy always stays at the current state; otherwise, for $\theta < 0$, the greedy policy always chooses to switch states. Interestingly, as in [1], this limited basis is still sufficient to express the optimal Q-function via $\theta = 0$ (there are no rewards).

We will show that the behaviour of approximate Q-learning with $\varepsilon$-greedy exploration can still diverge due to delusional bias; in particular that, after initializing to $\theta_0 = 1$, $\theta$ grows positively without bound. To do so, we examine the expected behaviour of the approximate Q-learning update under the stationary visitation frequencies. Note that, since $\theta > 0$ will hold throughout the analysis, the $\varepsilon$-greedy policy chooses to stay at the same state with probability $1 - \varepsilon$ and switches states with probability $\varepsilon$. Therefore, the stationary visitation frequencies, $\mu(s, a)$, is given by $\mu(s_1, a_1) = (1 - \varepsilon)/2$; $\mu(s_1, a_2) = \varepsilon/2$; $\mu(s_2, a_1) = \varepsilon/2$; and $\mu(s_2, a_2) = (1 - \varepsilon)/2$.

Consider the learning update, $\theta \leftarrow \theta + \alpha\Delta\theta$, where the *expected* update is given by

$$\mathbb{E}[\Delta\theta] = \sum_{s,a} \mu(s, a)\phi(s, a)\delta(s, a),$$

using the Q-learning temporal difference error, $\delta$, given by

$$\delta(s, a) = \gamma \sum_{s'} p(s'|s, a) \max_{a'} \phi(s', a')\theta - \phi(s, a)\theta$$

(recall the rewards are 0). In this example, the temporal differences are $\delta(s_1, a_1) = -(1 - \gamma)\theta/Z$; $\delta(s_1, a_2) = (3\gamma - 1)\theta/Z$; $\delta(s_2, a_1) = -(1 - \gamma)\theta/Z$; and $\delta(s_2, a_2) = -3(1 - \gamma)\theta/Z$, in the limit when $\eta \to 0$. Observe that $\delta(s_1, a_2)$ demonstrates a large delusional bias in this case (whenever $\theta > 0$). In particular, the other $\delta$ values are small negative numbers (assuming $\gamma \approx 1$), while $\delta(s_1, a_2)$ is close to $2/Z$. The delusion occurs because the update through $(s_1, a_2)$ thinks that a large future value can be obtained by switching to $s_2$, but the greedy policy allows no such switch. In particular, $\mathbb{E}[\Delta\theta]$ can be explicitly computed to be

$$\mathbb{E}[\Delta\theta] = \left(\frac{(5 - 3\varepsilon)\theta}{Z}\right)\gamma - \left(\frac{(5 - 4\varepsilon)\theta}{Z}\right),$$

which is bounded above zero for all $\theta \geq 1$ whenever $\delta > (5 - 4\varepsilon)/(5 - 3\varepsilon)$. Note that, since $\phi > 0$, $\delta$ is positive homogeneous in $\theta$. Therefore, the expected value of $\theta_k$ is

$$
\begin{aligned}
\mathbb{E}[\theta_k] &= \mathbb{E}[\theta_{k-1} + \alpha\,\mathbb{E}[\Delta\theta_{k-1}]] \\
&= \mathbb{E}[\theta_{k-1} + \alpha\theta_{k-1}\,\mathbb{E}[\Delta\theta_0]] \\
&= (1 + \alpha\,\mathbb{E}[\Delta\theta_0])\,\mathbb{E}[\theta_{k-1}],
\end{aligned}
$$

leading to divergence w.p. 1 (see, e.g., [10, Chap.4]) since we have established $\mathbb{E}[\Delta\theta_0] > 0$ for some $\gamma < 1$.

## A.3 Q-learning cyclic behaviour due to delusion

We show that delusion caused by the online Q-learning backup along an infeasible path leads to cycling of solutions and hence does not converge when the learning rate $\alpha_t^{sa}$ is lower bounded by $\alpha > 0$ (i.e. some schedule where $\alpha$ is the smallest learning rate). Consider Fig. 4. The path $(s_1, a_2)$, $(s_2, a_2)$ is infeasible: as it requires that $\theta_2 > 0$ (choosing $a_2$ over $a_1$ at $s_1$) and $\theta_2 < 0$ (taking $a_2$ at $s_2$). We assume $\gamma = 1$ for this episodic MDP.

There are four potential updates, each at a different $(s, a)$-pair. Any backup at $(s_1, a_1)$ or $(s_2, a_1)$ does not change the weight vector. Consider the first update at $(s_2, a_2)$ at iteration $k$ such that $\alpha_k = \alpha$. Assume $\theta^{(k-1)} = (b, c)$. A reward of $R(s_2, a_2) = 1/\sqrt{\alpha}$ is obtained, hence we have the update

$$\theta^{(k)} = (b, c) + \alpha \cdot \left(0, -\frac{1}{\sqrt{\alpha}}\right) \cdot \left(\frac{1}{\sqrt{\alpha}} + \gamma \cdot 0 - \left(-\frac{c}{\sqrt{\alpha}}\right)\right) = (b, -1) \ .$$

The next backup with non-zero updates occurs at $(s_1, a_2)$ at step $k'$ (since an update at $(s_2, a_2)$ would not change $\theta$ and updates at $(s_1, a_1)$ and $(s_2, a_1)$ gets multiplied by all zero features), $\theta^{(k'-1)} = (b, -1)$ and

$$\theta^{(k')} = (b, -1) + \alpha \cdot \left(0, \frac{1}{\sqrt{\alpha}}\right) \cdot \left(0 + \gamma \cdot \frac{1}{\sqrt{\alpha}} - \left(-\frac{1}{\sqrt{\alpha}}\right)\right) = (b, 1) \ .$$

This shows the cyclic behaviour of Q-learning when the two jointly infeasible state-action pairs "undo" each others' updates. We can easily extend this small example to larger feature spaces with larger collections of infeasible state-action pairs.

Figure 4: This two state MDP has $\phi(s_1, a_2) = (0, 1/\sqrt{\alpha})$, $\phi(s_2, a_2) = (0, -1/\sqrt{\alpha})$, $\phi(s_1, a_1) = \phi(s_2, a_1) = (0, 0)$. There is a single non-zero reward $R(s_2, a_2) = 1/\sqrt{\alpha}$. All transitions are deterministic and only one is non-terminal: $s_1, a_2 \to s_2$.

### A.4 The discounting paradox

We first illustrate the discounting paradox with the simple MDP in Fig. 5. Delusional bias causes approximate QL to behave paradoxically: a policy trained with $\gamma = 1$ will be *worse* than a policy trained with $\gamma = 0$, even when evaluated non-myopically using $\gamma = 1$.[3]

We use a linear approximator

$$Q_\theta(s, a) = \theta_1 \phi(s) + \theta_2 \phi(a) + \theta_3, \tag{4}$$

with the feature embeddings $\phi(s_1) = 2$; $\phi(s_2) = 1$; $\phi(s_2') = 0$; $\phi(a_1) = -1$; and $\phi(a_2) = 1$. The two discounts $\gamma = 0$ and $\gamma = 1$ each give rise to different optimal parameters, inducing the corresponding greedy policies:

$$\hat{\theta}^{\gamma=0} = \mathsf{QL}(\gamma = 0, \theta^{(0)}, \varepsilon\mathsf{Greedy}), \qquad \hat{\pi}_0 = \mathsf{Greedy}(Q_{\hat{\theta}^{\gamma=0}}),$$
$$\hat{\theta}^{\gamma=1} = \mathsf{QL}(\gamma = 1, \theta'^{(0)}, \varepsilon\mathsf{Greedy}), \qquad \hat{\pi}_1 = \mathsf{Greedy}(Q_{\hat{\theta}^{\gamma=1}}).$$

For all $\varepsilon \le 1/2$ we will show that $V_{\gamma=1}^{\hat{\pi}_0} > V_{\gamma=1}^{\hat{\pi}_1}$.

The (greedy) policy class representable by the linear approximator, $G(\Theta) = \{\mathsf{Greedy}(Q_\theta) : \theta \in \mathbb{R}^3\} = \{\pi_{a_1}, \pi_{a_2}\}$, is extremely limited: if $\theta_2 < 0$, the greedy policy always takes $a_1$ (policy $\pi_{a_1}$), while if $\theta_2 > 0$, it always takes $a_2$ (policy $\pi_{a_2}$). When evaluating these policies using $\gamma = 1$, we find that $V_{\gamma=1}^{\pi_{a_2}} = 2 > 2 - \delta = V_{\gamma=1}^{\pi_{a_1}}$, hence the optimal policy in $G(\Theta)$ is $\pi_{a_2}$. By contrast, the *unconstrained* optimal policy $\pi^*$ takes $a_1$ in $s_1$ and $a_2$ in $s_2$. The paradox arises because, as we will see, the myopic learner ($\gamma = 0$) converges to the best representable policy $\hat{\pi}_0 = \pi_{a_2}$, whereas the non-myopic learner ($\gamma = 1$) converges to the worse policy $\hat{\pi}_1 = \pi_{a_1}$.

To prove this result, we characterize the fixed points, $\hat{\theta}^{\gamma=0}$ and $\hat{\theta}^{\gamma=1}$, produced by QL with $\gamma = 0$ and $\gamma = 1$, respectively. (Note that the initial parameters $\theta^{(0)}, \theta'^{(0)}$ do not influence this analysis.) Using

Figure 5: A deterministic MDP starting at state $s_1$ and terminating at $T$, $Q_\theta(T, a) = 0$. Directed edges are transitions of the form $a/r$ where $a$ is the action taken and $r$ the reward. Parameter $\delta > 0$.

Figure 6: Fixed points $\hat{\theta}_2^\gamma$ when training with $\gamma = 0, 1$; $\delta = 0.1$.

the behaviour policy $\pi_b = \varepsilon$Greedy, any fixed point $\hat{\theta}$ of QL must satisfy two conditions (see also Appendix A.1):

**(1)** The behaviour policy must not change with each update; if $\text{sgn}(\hat{\theta}_2) = -1$ (resp. $+1$) then $\varepsilon$Greedy takes $a_1$ with probability $1 - \varepsilon$ (resp. $a_2$).

**(2)** The expected update values for $\theta$ must be zero under the stationary visitation frequencies $\mu(s, a)$ of the behaviour policy.[4]

The set of fixed points is entirely characterized by the sign of $\hat{\theta}_2$. For fixed points where $\hat{\theta}_2 < 0$ the second condition imposes the following set of constraints,

$$\sum_{s,a} \mu_{a_1}(s,a)\phi(s,a)\Big(R(s,a) + (\gamma s' - s)\hat{\theta}_1 - (\gamma + a)\hat{\theta}_2 + (\gamma - 1)\hat{\theta}_3\Big) = \mathbf{0} ; \quad \hat{\theta}_2 < 0 , \qquad (5)$$

where $s'$ follows $s, a$. Similarly, when $\hat{\theta}_2 > 0$, we have

$$\sum_{s,a} \mu_{a_2}(s,a)\phi(s,a)\Big(R(s,a) + (\gamma s' - s)\hat{\theta}_1 + (\gamma - a)\hat{\theta}_2 + (\gamma - 1)\hat{\theta}_3\Big) = \mathbf{0} ; \quad \hat{\theta}_2 > 0 . \qquad (6)$$

Note that there is no fixed point where $\hat{\theta}_2 = 0$.

If we set $\delta = 0.1$, $\varepsilon = 0.1$ and $\gamma = 1$ and solve for (5), we obtain $\hat{\theta}^{\gamma=1} \approx (0.793, -0.128, 0.265)$, whereas there is no solution for (6). Alternatively, by setting $\gamma = 0$, there is a solution to (6) given by $\hat{\theta}^{\gamma=0} \approx (-0.005, 0.03, 0.986)$, but there is no solution to (5). (Fig. 6 shows the entire family of fixed points as $\varepsilon$ varies.) Thus, $\text{Greedy}(Q_{\hat{\theta}^{\gamma=1}})$ (the non-myopically trained policy) always takes action $a_1$, whereas $\text{Greedy}(Q_{\hat{\theta}^{\gamma=0}})$ (the myopically trained policy) always takes action $a_2$. When evaluating these policies under $\gamma = 1$, the myopic policy has value 2, while paradoxically the non-myopic policy has lower value $2 - \delta$.

This discounting paradox arises precisely because Q-learning fails to account for the fact that the class of (greedy) policies admitted by the simple linear approximator is extremely limited. In particular, under non-myopic ($\gamma = 1$) training, QL "believes" that taking $a_1$ in $s_1$ results in a better long-term reward, which is in fact true if we can execute the *unconstrained* optimal policy: take $a_1$, then $a_2$ (i.e., the top path), receiving a total reward of $2 + \delta$. However, this policy is infeasible, since the

Figure 7: An MDP similar to Fig. 5; non-myopic is much worse.

policy class $G(\Theta)$ consists of greedy policies that can only take $a_1$ or $a_2$ at all states: if $a_1$ is taken at $s_1$, it must also be taken at $s_2$.

More specifically, the paradox emerges because, in the temporal difference, the $\gamma \max_{a' \in A} Q(s', a')$ term maximizes over all possible actions, without regard to actions that may have been taken to reach $s'$. If reaching $s'$ requires taking some prior action that renders a specific $a'$ infeasible at $s'$ w.r.t. $G(\Theta)$, then Q-learning "deludes" itself, believing it can achieve future values that are, in fact, infeasible under the constrained policy class $G(\Theta)$.

The example above illustrates one instance of the discounting paradox, showing that training using the larger correct discount factor can give rise to a slightly worse policy than training using an "incorrect" smaller discount. We now consider how bad the performance loss can be and also show that the paradox can occur in the opposite direction (smaller target discount outperformed by larger incorrect discount).

The two induced greedy policies above, $\pi_{a_1}$ and $\pi_{a_2}$, have a difference in value $\delta$ (see Fig. 5) when evaluated with $\gamma = 1$; $\delta$ must be relatively small for the paradox to arise (e.g., if $\varepsilon = 0.1$, $\delta \lessgtr 0.23$). A different MDP (see Fig. 7) induces a delusional bias in which the non-myopic Q-learned policy can be *arbitrarily worse* (in the competitive-ratio sense) than the myopic Q-learned policy. To show this we use the same linear approximator (4) and action embeddings, but new feature embeddings $\phi(s_1) = 3$, $\phi(s_2) = 2$, $\phi(s_2') = 1.9999$, $\phi(s_3) = 1.1$, and $\phi(s_3') = 1.0999$. When evaluating with $\gamma = 1$, the policy $\pi_{a_2}$ (with value 3) has an advantage of $0.1\delta$ over $\pi_{a_1}$ (with value $3 - 0.1\delta$). Assume the behaviour policy is $\varepsilon$Greedy with $\varepsilon = 0.2$ (other $\varepsilon$ also work), whose distribution $\mu(s, a)$ can be computed as in Footnote 4.

For Q-learning with $\gamma = 1$, we solve the constraints in (5) to find fixed points where $\hat{\theta}_2 < 0$. The system of equations has the form $\mathbf{A}\hat{\theta} = \mathbf{b}$; solving for $\hat{\theta}$ gives:

$$\hat{\theta}^{\gamma=1} = \mathbf{A}^{-1}\mathbf{b} \approx \begin{bmatrix} 1.03320 + 0.09723\delta \\ -0.00002 - 0.08667\delta \\ -0.09960 - 0.65835\delta \end{bmatrix} .$$

This implies, for any $\delta \geq 0$, the Q-learned greedy policy is $\pi_{a_1}$. Similarly, fixed points with $\gamma = 0$, where $\hat{\theta}_2 > 0$, we have $\hat{\theta}_2^{\gamma=0} \approx 0.10333\delta$. This implies the Q-learned greedy policy is $\pi_{a_2}$ for any $\delta > 0$. In particular, we have that

$$\lim_{\delta \to 30^-} \frac{V_{\gamma=1}^{\mathsf{Greedy}(\hat{\theta}^{\gamma=1})}}{V_{\gamma=1}^{\mathsf{Greedy}(\hat{\theta}^{\gamma=0})}} = \frac{3 - 0.1(30)}{3} = 0 .$$

Therefore the non-myopic policy may be arbitrarily worse than the myopic policy. (Other fixed points, where $\hat{\theta}_2^{\gamma=1} > 0, \hat{\theta}_2^{\gamma=0} < 0$ may be reached depending on the initial $\theta^{(0)}$.)

The paradox can also arise in the "opposite" direction, when evaluating policies purely myopically with $\gamma = 0$: the myopic Q-learned policy can be arbitrarily worse than a non-myopic policy. As shown above, for any $\delta > 0$, the non-myopic policy chooses $a_1$ at $s_1$, thus its value is $1 + \delta$ (since subsequent rewards are fully discounted) whereas the myopic policy chooses $a_2$ receiving a value of 1. Thus, the non-myopic policy has a value advantage of $\delta$, which translates into an arbitrarily large improvement ratio over the myopic policy: $\lim_{\delta \to \infty}(1 + \delta)/1 = \infty$.

## A.5 Comparisons to double Q-learning

The maximization bias in Q-learning [30] is an over-estimation bias of the bootstrap term $\max_{a' \in A} \hat{Q}(s', a')$ in Q-updates. If the action space is large, and there are not enough transition examples to confidently estimate the true $Q(s', a')$, then the variance of each $\hat{Q}(s, a)$ for any $a$ is large. Taking the max of random variables with high variance will over-estimate, even if $\max_{a' \in A} \mathbb{E}[\hat{Q}(s, a)] = \max_{a' \in A} Q(s, a)$.

Double Q-learning aim to fix the maximization bias by randomly choosing to update one of two Q-functions, $Q_1$ and $Q_2$. For any given transition $(s, a, r, s)$, if $Q_1$ is chosen to be updated with probability $1/2$, then the update becomes

$$Q_1(s, a) \leftarrow Q_1(s, a) + \alpha(r + \gamma Q_2(\operatorname*{argmax}_{a' \in A} Q_1(s', a') - Q_1(s, a))\nabla Q_1(s, a)$$

If $Q_2$ is randomly chosen, a similar update is applied by switching the roles of $Q_1$ and $Q_2$. A common behaviour policy is $\varepsilon\mathsf{Greedy}(Q_1 + Q_2)$. The idea behind this update is to reduce bias by having $Q_2$, trained with different transitions, evaluate the max action of $Q_1$.

This type of maximization bias is due to lack of samples, which causes large variance that biases the max value. The MDP counter-example in Fig. 5 is deterministic, both in the rewards and transitions, so in fact there is no maximization bias. To make this concrete, we can derive the double Q-learning fixed points $Q_\theta$ and $Q_{\theta'}$ of Fig. 5. Using similar reasoning as in Eq. 5, we can write the constraints for fixed points where $\theta_2 + \theta_2' < 0$ (resp. $> 0$)—i.e. Greedy always selects $a_1$ (resp. always $a_2$).

$$\sum_{s,a} \mu(s, a)\phi(s, a)(R(s, a) + \gamma(\theta_1 s' + \theta_2 a' + \theta_3) - \theta_1' s - \theta_2' a - \theta_3') = \mathbf{0} \tag{7}$$

$$\sum_{s,a} \mu(s, a)\phi(s, a)(R(s, a) + \gamma(\theta_1' s' + \theta_2' a'' + \theta_3') - \theta_1 s - \theta_2 a - \theta_3) = \mathbf{0} \tag{8}$$

The weighting $\mu(s, a) = \frac{1}{2}\mu_b(s, a)$ where $\mu_b(s, a)$ corresponds to the behaviour policy. If $\theta_2 + \theta_2' < 0$, $\varepsilon\mathsf{Greedy}$ prefers $a_1$, and there are one of three possibilities. First, $a' = a'' = a_1$ implying $\theta_2, \theta_2' < 0$; second, $(a', a'') = (a_1, a_2)$ implying $\theta_2 > 0, \theta_2' < 0$; third, $(a', a'') = (a_2, a_1)$ implying $\theta_2 < 0, \theta_2' > 0$. We can substitute all three cases into Eqs. 7, 8, solve the system of equations, and check if $\theta, \theta'$ satisfy the constraints. One solution is a fixed point of regular Q-learning where $Q_\theta = Q_\theta'$. For both $\gamma = 0, 1$, we check for other possible solutions with the MDP of Fig 5, they do not exist. Fixed points when $\theta_2 + \theta_2' > 0$ can be found similarly. Thus double Q-learning does not mitigate the delusional bias issue.

## A.6 Concepts and proofs for PCVI and PCQL

In this section, we elaborate on various definitions and provide proofs of the results in Section 4.

We formally define functions and binary operators acting on partitions of $\Theta$.

**Definition 4.** *Let $\mathcal{X}$ be a set, a* finite partition *of $\mathcal{X}$ is any set of non-empty subsets $P = \{X_1, \ldots, X_k\}$ such that $X_1 \cup \cdots \cup X_k = \mathcal{X}$ and $X_i \cap X_j = \emptyset$, for all $i \neq j$. We call any $X_i \in P$ a* cell. *A partition $P'$ of $\mathcal{X}$ is a* refinement *of $P$ if for all $X' \in P'$ there exists a $X \in P$ such that $X' \subseteq X$. Let $\mathcal{P}(\mathcal{X})$ denote the set of all finite partitions of $\mathcal{X}$.*

**Definition 5.** *Let $P \in \mathcal{P}(\mathcal{X})$. A mapping $h : P \to \mathbb{R}$ is called a* function of partition $P$. *Let $\mathcal{H} = \{h : P \to \mathbb{R} \mid P \in \mathcal{P}(\mathcal{X})\}$ be the set of all such functions of partitions. Let $h_1, h_2 \in \mathcal{H}$, an* intersection sum *is a binary operator $h = h_1 \oplus h_2$ defined by*

$$h(X_1 \cap X_2) = h_1(X_1) + h_2(X_2), \quad \forall X_1 \in \mathsf{dom}(h_1), X_2 \in \mathsf{dom}(h_2), X_1 \cap X_2 \neq \emptyset$$

*where $\mathsf{dom}(\cdot)$ is the domain of a function (in this case a partition of $\mathcal{X}$). We say $h_1$ is a* refinement *of $h_2$ if partition $\mathsf{dom}(h_1)$ is a refinement of $\mathsf{dom}(h_2)$.*

Note there is at most a quadratic blowup: $|\mathsf{dom}(h)| \leq |\mathsf{dom}(h_1)| \cdot |\mathsf{dom}(h_2)|$. The tuple $(\mathcal{H}, \oplus)$ is almost an abelian group, but without the inverse element property.

**Proposition 6.** *The following properties hold for the intersection sum.*

- *(Identity) Let $e = X \mapsto 0$, then $h \oplus e = e \oplus h = h$ for all $h \in \mathcal{H}$.*

- *(Refinement) For all $h_1, h_2 \in \mathcal{H}$, we have: (i) $h_1 \oplus h_2$ is a refinement of $h_1$ and of $h_2$. (ii) if $h_1'$ and $h_2'$ is a refinement of $h_1$ and $h_2$, respectively, then $h_1' \oplus h_2'$ is a refinement of $h_1 \oplus h_2$.*

- *(Closure) $h_1 \oplus h_2 \in \mathcal{H}$, for all $h_1, h_2 \in \mathcal{H}$*

- *(Commutative) $h_1 \oplus h_2 = h_2 \oplus h_1$, for all $h_1, h_2 \in \mathcal{H}$*

- *(Associative) $(h_1 \oplus h_2) \oplus h_3 = h_1 \oplus (h_2 \oplus h_3)$, for all $h_1, h_2, h_3 \in \mathcal{H}$*

*Proof.* The identity element $e$ property follows trivially. Refinement (i) follows from the fact that for any two partitions $P_1, P_2 \in \mathcal{P}(X)$, say $P_1 = \mathsf{dom}(h_1)$ and $P_2 = \mathsf{dom}(h_2)$, the set $P = \mathsf{dom}(h_1 \oplus h_2) = \{X_1 \cap X_2 \mid X_1 \in P_1, X_2 \in P_2, X_1 \cap X_2 \neq \emptyset\}$ is also a partition where $X_1 \cap X_2 \in P$ is a subset of both $X_1 \in P_1$ and $X_2 \in P_2$. Refinement (ii) is straightforward, let $X' \in \mathsf{dom}(h_1')$ and $Y' \in \mathsf{dom}(h_2')$ where there is a non-empty intersection. By definition, $X' \subseteq X \in \mathsf{dom}(h_1)$ and $Y' \subseteq Y \in \mathsf{dom}(h_2)$. We have $X' \cap Y' \in \mathsf{dom}(h_1' \oplus h_2')$ and $(X' \cap Y') \subseteq (X \cap Y)$, so refinement (ii) holds. Commutative and associative properties essentially follow from that of the corresponding properties of set intersection and the addition operators. Closure follows from the refinement property since $h_1 \oplus h_2$ is a mapping whose domain is the refined (finite) partition. □

**Assumption 7.** *We have access to an oracle $\mathsf{Witness}$ where for any $X \subseteq \Theta$ defined by a conjunction of a collection of state to action constraints, outputs $\emptyset$ if $X$ is an inconsistent set of constraints, and outputs any witness $\theta \in X$ otherwise.*

**Theorem 1.** *PCVI (Alg. 1) has the following guarantees:*

(a) *(Convergence and correctness) The $\mathbb{Q}$ function converges and, for each $s \in S, a \in A$, and any $\theta \in \Theta$: there is a unique $X \in \mathsf{dom}(\mathbb{Q}[sa])$ s.t. $\theta \in X$ and*

$$Q^{\pi_\theta}(s, a) = \mathbb{Q}[sa](X). \tag{9}$$

(b) *(Optimality and Non-delusion) Given initial state $s_0$, $\pi_{\theta^*}$ is an optimal policy within $G(\Theta)$ and $q^*$ is its value.*

(c) *(Runtime bound) Assume $\oplus$ and non-emptiness checks (lines 6 and 7) have access to $\mathsf{Witness}$. Let*

$$\mathcal{G} = \{g_\theta(s, a, a') := \mathbf{1}[f_\theta(s, a) - f_\theta(s, a') > 0], \forall s, a \neq a' \mid \theta \in \Theta\}, \tag{10}$$

*where $\mathbf{1}[\cdot]$ is the indicator function. Then each iteration of Alg. 1 runs in time $O(nm \cdot [\binom{m}{2}n]^{2\,\mathsf{VCDim}(\mathcal{G})}(m-1)w)$ where $\mathsf{VCDim}(\cdot)$ is the VC-dimension [31] of a set of boolean-valued functions, and $w$ is the worst-case running time of the oracle called on at most $nm$ state-action constraints. Combined with Part (a), if $\mathsf{VCDim}(\mathcal{G})$ is finite then $\mathbb{Q}$ converges in time polynomial in $n, m, w$.*

**Corollary 8.** *Let $\phi : S \times A \to \mathbb{R}^d$ be a vector representation of state-action pairs. Define*

- *$\mathcal{F}_{\text{linear}} = \{f_\theta(s, a) = \theta^T \phi(s, a) + \theta_0 \mid \theta \in \mathbb{R}^d, \theta_0 \in \mathbb{R}\}$ and*

- *$\mathcal{F}_{\text{DNN}}$ the class of real-valued ReLU neural networks with input $\phi(s, a)$, identity output activation, $W$ the number of weight parameters, and $L$ the number of layers.*

*Let $\mathcal{G}_{\text{linear}}$ and $\mathcal{G}_{\text{DNN}}$ be the corresponding boolean-valued function class as in Eqn. 10. Then*

- *$\mathsf{VCDim}(\mathcal{G}_{\text{linear}}) = d$ and*

- *$\mathsf{VCDim}(\mathcal{G}_{\text{DNN}}) = O(WL \log W)$.*

*Furthermore, the $\mathsf{Witness}$ oracle can be implemented in polynomial time for $\mathcal{F}_{\text{linear}}$ but is NP-hard for neural networks. Therefore Alg. 1 runs in polynomial time for linear greedy policies and runs in polynomial time as a function of the number of oracle calls for deep Q-network (DQN) greedy policies.*

*Proof.* Let $g_\theta \in \mathcal{G}_{\text{linear}}$ then $g_\theta(s, a, a') = \mathbf{1}[\theta^T(\phi(s, a) - \phi(s, a')) > 0]$, these linear functions have VC-dimension $d$. For any $f_\theta \in \mathcal{F}_{\text{DNN}}$ we construct a ReLU network $g_\theta$ as follows. The first input $\phi(s, a)$ goes through the network $f_\theta$ with identity output activation and the second input $\phi(s, a')$ goes through the same network, they are then combined with the difference $f_\theta(s, a) - f_\theta(s, a')$ before

being passed through the $\mathbf{1}[\cdot]$ output activation function. Such as network has $2W$ number of weight parameters (with redundancy) and the same $L$ number of layers. Then the VC-dimension of this network follows from bounds given in [2].

The oracle can be implemented for linear policies by formulating it as set of linear inequalities and solving with linear programming methods. For neural networks, the problem of deciding if a training set can be correctly classified is a classical NP-hardness result [7] and can be reduced to this oracle problem by converting the training data into state to action constraints. $\qquad\square$

Before proving the main theorem, we first show that the number of unique policies is polynomial in $n$ when the VC-dimension is finite. This implies a bound on the blowup in the number of cells under the $\oplus$ operator.

**Proposition 9.** *Let $\mathcal{G}$ be defined as in Eqn. 10, then we have*

$$|G(\Theta)| \leq \sum_{i=0}^{\mathsf{VCDim}(\mathcal{G})} \binom{\binom{m}{2}n}{i} = O\left(\left[\binom{m}{2}n\right]^{\mathsf{VCDim}(\mathcal{G})}\right).$$

*Proof.* We construct a one-to-one mapping from functions in $G(\Theta)$ to functions in $\mathcal{G}$. Let $\pi_\theta \in G(\Theta)$. Suppose $\pi_\theta(s) = a$ for some state $s$. This implies $f_\theta(s,a) > f_\theta(s,a')$ for all $a' \neq a$, or equivalently $\mathbf{1}[f_\theta(s,a) - f_\theta(s,a') > 0] = 1$ for all $a' \neq a$. The converse is also true by definition. Thus the mapping $\pi_\theta \mapsto g_\theta$ is one-to-one, but not necessarily onto since $g_\theta$ is sensitive to all pairwise comparisons of $a$ and $a'$. For ties we can assume both $f$ and $g$ selects the action with smallest index in $A$). The Sauer-Shelah Lemma [24, 25] gives us the bound $|\mathcal{G}| \leq \sum_{i=0}^{\mathsf{VCDim}(\mathcal{G})} \binom{|\mathsf{dom}(\mathcal{G})|}{i}$ where $|\mathsf{dom}(\mathcal{G})| = |S \times A \times A - \{(a,a) \mid a \in A\}| = \binom{m}{2}n$. $\qquad\square$

*Proof of Theorem 1.* **Part (a):** first, let us argue that for any state-action pair $s, a$, these two conditions hold when executing the algorithm,

(1) $\mathsf{dom}(\mathtt{Q}[sa])$ is always a partition of $\Theta$ and

(2) $\mathsf{dom}(\mathtt{ConQ}[sa])$ is always a partition of $[s \mapsto a]$.

Note that the initialization in lines 1 and 2 satisfy these two conditions. Consider any iteration of the algorithm. We have that $\mathsf{dom}(\mathtt{ConQ}[s])$ is a partition of $\Theta$ since $\mathsf{dom}(\mathtt{ConQ}[s]) = \dot{\bigcup}_{a \in A} \mathsf{dom}(\mathtt{ConQ}[sa])$ and each $\mathsf{dom}(\mathtt{ConQ}[sa])$ is a partition of $[s \mapsto a]$ (this is the invariant condition in the loop). Hence $\mathtt{ConQ}[s]$ is a function of a partition of $\Theta$. Line 6 is an update that is well-defined and results in $\mathsf{dom}(\mathtt{Q}[sa]) \in \mathcal{P}(\Theta)$, this is due to the commutative, associative and closure properties of the intersection sum (see Prop. 6). Thus condition (1) is satisfied. The update in line (7) ensures that $\mathsf{dom}(\mathtt{ConQ}[sa])$ has no empty sets and that each set $Z$ in its domain comes from $\mathsf{dom}(\mathtt{Q}[sa])$ but with the constraint that $\pi_\theta(s) = a$. Thus condition (2) is satisfied.

To show convergence of $\mathtt{Q}$, we first show the partitions $\mathsf{dom}(\mathtt{ConQ}[s]) \in \mathcal{P}(\Theta)$ eventually converges—i.e. they do not change. Since partitions in $\mathtt{Q}$ is derived from partitions in $\mathtt{ConQ}$, then $\mathtt{Q}$ would also converge. Once partitions converge, we show that the backups in line 6 contain value iteration updates for policies within each cell. An application of standard convergence rates for value iteration gives us the desired results.

We first show that after each iteration of the inner loop, $\mathtt{ConQ}[s]$ is a refinement of the old $\mathtt{ConQ}[s]$ of a previous iteration. Let $\mathtt{Q}^{(i)}, \mathtt{ConQ}^{(i)}$, denote the tables at iteration $i$ of the outer loop—that is, after executing $i$ full passes of the inner loop. We claim that $\mathtt{ConQ}^{(i)}[s]$ is a refinement of $\mathtt{ConQ}^{(i-1)}[s]$ for all $s$. We prove this by induction on $i$. Base case: $i = 1$; $\mathtt{ConQ}^{(0)}[s]$ are partitions of the form $\{[s \mapsto a_1], \ldots, [s \mapsto a_m]\}$. Line 7 ensures $\mathtt{ConQ}^{(1)}[sa]$ is a refinement of $[s \mapsto a]$ since $\mathtt{Q}[sa]$ is a partition of $\Theta$ (shown above). Since this is true for any $a$ then $\mathtt{ConQ}^{(1)}[s]$ is a refinement of $\mathtt{ConQ}^{(0)}[s]$. For $i > 1$, line 6 ensures that $\mathsf{dom}(\mathtt{Q}^{(i)}[sa]) = \mathsf{dom}(\bigoplus_{s'} \mathtt{ConQ}^{(i-1)}[s'])$ and $\mathsf{dom}(\mathtt{Q}^{(i-1)}[sa]) = \mathsf{dom}(\bigoplus_{s'} \mathtt{ConQ}^{(i-2)}[s'])$. By inductive hypothesis $\mathtt{ConQ}^{(i-1)}[s']$ is a refinement of $\mathtt{ConQ}^{(i-2)}[s']$. Therefore by Prop. 6's refinement property, $\mathtt{Q}^{(i)}[sa]$ is a refinement of $\mathtt{Q}^{(i-1)}[sa]$. By line 7 it follows that $\mathtt{ConQ}^{(i)}[sa]$ is a refinement of $\mathtt{ConQ}^{(i-1)}[sa]$. Finally $\mathtt{ConQ}^{(i)}[s]$ is the combination of all such relevant refinements, resulting in a refinement of $\mathtt{ConQ}^{(i-1)}[s]$. This ends the induction proof.

Thus a full pass of the inner loop may result in new cells for $\mathtt{ConQ}^{(i)}[s]$ (part of a refinement) but never reduce the number of cells (i.e. never be an "anti-refinement"). If no new cells in $\mathtt{ConQ}^{(i)}[s]$ are introduced in a full pass of the inner loop, i.e. $\mathrm{dom}(\mathtt{ConQ}^{(i)}[s]) = \mathrm{dom}(\mathtt{ConQ}^{(i-1)}[s])$ for all $s$, then $\mathrm{dom}(\mathtt{Q})$ (and hence $\mathrm{dom}(\mathtt{ConQ}[s])$) has converged. To see this, note that $\mathrm{dom}(\mathtt{Q}^{(i+1)}[sa]) = \mathrm{dom}(\bigoplus_{s'} \mathtt{ConQ}^{(i)}[s']) = \mathrm{dom}(\bigoplus_{s'} \mathtt{ConQ}^{(i-1)}[s']) = \mathrm{dom}(\mathtt{ConQ}^{(i)}[s])$. This implies $\mathrm{dom}(\mathtt{ConQ}^{(i+1)}[s]) = \mathrm{dom}(\mathtt{ConQ}^{(i)}[s])$ for all $s$. Hence in all subsequent iterations the partitions in $\mathtt{Q}$ and $\mathtt{ConQ}$ does not change, i.e. it converges. The maximum number of full passes of the inner loop before no new cells are introduced in a full inner pass is the maximum total number of cells in $\mathtt{ConQ}[s]$ summed across all states (in the worst case when each full pass of inner loop results in only one new cell), since this maximum total is bounded, $\mathtt{Q}$ converges eventually.

Assume that cells in $\mathtt{Q}$ and $\mathtt{ConQ}$ has converged. Let $\theta \in \Theta$, now we will show the update in line 6 contains the update

$$Q^{\pi_\theta}(s, a) \leftarrow R_{sa} + \gamma \sum_{s' \in S} p(s' \mid s, a) Q^{\pi_\theta}(s', \pi(s')).$$

This is the on-policy value iteration update, which converges to the Q-function of $\pi_\theta$ with enough updates, thus Eqn. (9) would hold. Fix $s, a$. For any next state $s'$, there exists a $X_{s'} \in \mathrm{dom}(\mathtt{ConQ}[s'])$ such that $\theta \in X_{s'}$ (as we've established above that $\mathrm{dom}(\mathtt{ConQ}[s']) \in \mathcal{P}(\Theta)$). Moreover, let $Y = \bigcap_{s' \in S} X_{s'}$, then by definition $\theta \in Y \in \mathrm{dom}(\mathtt{Q}[sa])$. Thus the update in line 6 has the intersection sum where sets $X_{s'}$, for all $s'$, are combined through an intersection, that is, it contains the update

$$\mathtt{Q}[sa](Y) \leftarrow R_{sa} + \gamma \sum_{s' \in S} p(s' \mid s, a) \mathtt{ConQ}[s'](X_{s'}),$$

where $\mathtt{ConQ}[s'](X_{s'})$ stores the backed-up Q-value of $\theta$ in state $s'$ performing action $\pi_\theta(s')$, the consistent action for $\theta$. Line 7 ensures that only $\mathtt{ConQ}[s\ \pi_\theta(s)]$ contains the cell $Z' = Y' \cap [s \mapsto \pi_\theta(s)]$ containing parameter $\theta$ (all other $\mathtt{ConQ}[sa']$ does not contain a cell that contains $\theta$, for $a' \neq \pi_\theta(s)$). Thus, only $\mathtt{ConQ}[s\ \pi_\theta(s)](Z')$ is updated with $\mathtt{Q}[s\ \pi_\theta(s)](Y')$ which is consistent with $\theta$'s action in state $s$. Hence, we've established that for any $\pi_\theta$ its corresponding Q-values is updated at every iteration and stored in $\mathtt{Q}$ while only the consistent Q-values are stored in $\mathtt{ConQ}$.

**Part (b):** by construction and from above results, $\mathrm{dom}(\mathtt{ConQ}[s_0]) \in \mathcal{P}(\Theta)$. By Part (a), for any policy $\pi_\theta$, there exists a $X_\theta$ such that the policy has value $\mathtt{Q}[s_0\ \pi_\theta(s_0)](X_\theta) = \mathtt{ConQ}[s_0\ \pi_\theta(s_0)](X_\theta \cap [s_0 \mapsto \pi_\theta(s_0)]) = \mathtt{ConQ}[s_0](X_\theta \cap [s_0 \mapsto \pi_\theta(s_0)])$ when starting from $s_0$. Thus $q^* = \mathtt{ConQ}[s_0][X^*] \geq \mathtt{ConQ}[s_0][X_\theta]$ is the largest Q-value of any policy in $G(\Theta)$, upon convergence of $\mathtt{Q}$. The oracle returns a particular witness in $X^*$ whose greedy policy attains value $q^*$.

**Part (c):** line 6 dominates the running time. There are $m - 1$ applications of $\oplus$ operator, which is implemented by intersecting all pairs of cells from two partitions. There is a call to Witness to determine if the intersection of any pair of cells results in an inconsistent set of constraints, taking time at most $w$. Prop. 9 upper bounds the number of cells in a partition, thus we require $O([\binom{m}{2}]^{\mathsf{VCDim}\,\mathcal{G}} n\ (m-1)w)$ time for line 6, which is executed $nm$ times within the inner loop. The number of iterations until $\mathrm{dom}(\mathtt{Q}[s])$ converges is the maximum number of cells in $\mathtt{ConQ}[s]$ summed across all states $s$ (shown in proof of Part (a)). For finite VC-dimension, this summed total is polynomial. Once the partitions of $\mathtt{Q}[s]$ converges, standard results concerning polynomial time convergence of Q-values within each cell apply. $\qquad\square$

### A.7 PCVI example

We walk through the steps of the PCVI Algorithm for the example MDP in Fig. 1 and show how it computes the optimal admissible policy. We assume the same feature representation as in the discussion of Sec. 3.1; see Table 3 for the features and rewards.

Recall that the optimal admissible policy has parameters $\theta^* = (-2, 0.5)$, and its greedy policy $\pi_{\theta^*}$ selects $a_1$ at $s_1$ and $a_2$ at $s_4$, giving expected value of $0.5$ at initial state $s_1$. We walk through each step of PCVL below. Table 4 describes the critical data structures used during PCVL updates.

Table 3: Features and rewards for MDP in Fig. 1. Transitions are as in Fig. 1 with a stochastic transition at $s_1, a_1$ which goes to $s_4$ with probability $0.1$ and terminates with probability $0.9$. Initial state is $s_1$ and $\gamma = 1$.

| $s, a$ | $\phi(s,a)$ | $R(s,a)$ | $s, a$ | $\phi(s,a)$ | $R(s,a)$ |
|---|---|---|---|---|---|
| $s_1, a_1$ | $(0,1)$ | $0.3$ | $s_3, a_1$ | $(0,0)$ | $0$ |
| $s_1, a_2$ | $(0.8,0)$ | $0$ | $s_3, a_2$ | $(-1,0)$ | $0$ |
| $s_2, a_1$ | $(0,0)$ | $0$ | $s_4, a_1$ | $(0,1)$ | $0$ |
| $s_2, a_2$ | $(0.8,0)$ | $0$ | $s_4, a_2$ | $(-1,0)$ | $2$ |

Table 4: Explanation of critical data structures used in PCVL algorithm.

| Data structure | Description |
|---|---|
| Q[$sa$] | A table mapping an element (set) $X$ of a partition of $\Theta$ to $\mathbb{R}$. Q[$sa$]($X$) is the Q-value of taking action $a$ at $s$ and then following a greedy policy parameterized by $\theta \in X$. It *may be* that $\pi_\theta(s) \neq a$. |
| ConQ[$sa$] | A table mapping an element (set) $X$ of a partition of $[s \to a] = \{\theta \in \Theta : \pi_\theta(s) = a\}$ to $\mathbb{R}$. ConQ[$sa$]($X$) is the Q-value of taking action $a$ at $s$ and then following a greedy policy parameterized by $\theta \in X$. It *must be* that $\pi_\theta(s) = a$. ConQ is short for consistent Q-values. |
| ConQ[$s$] | A table formed from concatenating tables ConQ[$sa$] for all $a$ at $s$. The domain of ConQ[$s$] is a partition of $\Theta$. |

**Initialization.** First initialize Q and ConQ tables for all $i \in \{1, 2, 3, 4\}$ and $j \in \{1, 2\}$.

$$\text{Q}[s_i a_j] = \left[ \begin{array}{c|c} \text{Partition of } \Theta & \text{Q-values} \\ \hline \Theta & 0 \end{array} \right]$$

$$\text{ConQ}[s_i a_j] = \left[ \begin{array}{c|c} \text{Partition of } [s_i \mapsto a_j] & \text{Q-values} \\ \hline [s_i \mapsto a_j] & 0 \end{array} \right]$$

$$\text{ConQ}[s_i] = \left[ \begin{array}{c|c} \text{Partition of } \Theta & \text{Q-values} \\ \hline \Theta & 0 \end{array} \right]$$

Let $\perp$ be the terminal state and let ConQ[$\perp$] be initialized as above, in the same way as ConQ[$s_i$].

Figure 8: Example run of PCVI on MDP in Fig. 1. The blue, green, red and gray arrows indicate the direction of backups for constructing the tables ConQ[$s_i$] for $i = 1..4$ in that order.

**Iteration #1.** We now go through PCVL's updates within the for loop (lines 5-8). We run the updates backwards, starting with $s_4, a_2$. The updates are:

$$\mathbb{Q}[s_4 a_2] = R(s_4, a_2) + \gamma[p(\perp | s_4, a_2)\mathtt{ConQ}[\perp]]$$

$$= \begin{bmatrix} \text{Partition of } \Theta & \text{Q-values} \\ \hline \Theta & 2 \end{bmatrix}$$

$$\mathtt{ConQ}[s_4 a_2] = \begin{bmatrix} \text{Partition of } [s_4 \mapsto a_2] & \text{Q-values} \\ \hline \Theta \cap [s_4 \mapsto a_2] = [s_4 \mapsto a_2] & 2 \end{bmatrix}$$

$$\mathtt{ConQ}[s_4] = \begin{bmatrix} \text{Partition of } \Theta & \text{Q-values} \\ \hline [s_4 \mapsto a_1] & 0 \\ [s_4 \mapsto a_2] & 2 \end{bmatrix}$$

**Iteration #2.** Update $s_4, a_1$.

$$\mathbb{Q}[s_4 a_1] = R(s_4, a_1) + \gamma[p(\perp | s_4, a_1)\mathtt{ConQ}[\perp]]$$

$$= \begin{bmatrix} \text{Partition of } \Theta & \text{Q-values} \\ \hline \Theta & 0 \end{bmatrix}$$

$$\mathtt{ConQ}[s_4 a_2] = \begin{bmatrix} \text{Partition of } [s_4 \mapsto a_1] & \text{Q-values} \\ \hline \Theta \cap [s_4 \mapsto a_1] = [s_4 \mapsto a_1] & 0 \end{bmatrix}$$

$$\mathtt{ConQ}[s_4] = \begin{bmatrix} \text{Partition of } \Theta & \text{Q-values} \\ \hline [s_4 \mapsto a_1] & 0 \\ [s_4 \mapsto a_2] & 2 \end{bmatrix}$$

**Iteration #3.** Update $s_3, a_2$.

$$\mathbb{Q}[s_3 a_2] = R(s_3, a_2) + \gamma[p(s_4 | s_3, a_2)\mathtt{ConQ}[s_4]]$$

$$= \mathtt{ConQ}[s_4]$$

$$= \begin{bmatrix} \text{Partition of } \Theta & \text{Q-values} \\ \hline [s_4 \mapsto a_1] & 0 \\ [s_4 \mapsto a_2] & 2 \end{bmatrix}$$

$$\mathtt{ConQ}[s_3 a_2] = \begin{bmatrix} \text{Partition of } [s_3 \mapsto a_2] & \text{Q-values} \\ \hline [s_4 \mapsto a_1] \cap [s_3 \mapsto a_2] = [s_4 \mapsto a_1][s_3 \mapsto a_2] & 0 \\ [s_4 \mapsto a_2] \cap [s_3 \mapsto a_2] = [s_4 \mapsto a_2][s_3 \mapsto a_2] & 2 \end{bmatrix}$$

$$\mathtt{ConQ}[s_3] = \begin{bmatrix} \text{Partition of } \Theta & \text{Q-values} \\ \hline [s_3 \mapsto a_1] & 0 \\ [s_4 \mapsto a_1][s_3 \mapsto a_2] & 0 \\ [s_4 \mapsto a_2][s_3 \mapsto a_2] & 2 \end{bmatrix}$$

**Iteration #4.** Update $s_3, a_1$.

$$\mathbb{Q}[s_3 a_1] = R(s_3, a_1) + \gamma[p(\perp | s_3, a_1)\mathtt{ConQ}[\perp]]$$

$$= \mathtt{ConQ}[\perp]$$

$$= \begin{bmatrix} \text{Partition of } \Theta & \text{Q-values} \\ \hline \Theta & 0 \end{bmatrix}$$

$$\mathtt{ConQ}[s_3 a_1] = \begin{bmatrix} \text{Partition of } [s_3 \mapsto a_1] & \text{Q-values} \\ \hline \Theta \cap [s_3 \mapsto a_1] = [s_3 \mapsto a_1] & 0 \end{bmatrix}$$

$$\mathtt{ConQ}[s_3] = \begin{bmatrix} \text{Partition of } \Theta & \text{Q-values} \\ \hline [s_3 \mapsto a_1] & 0 \\ [s_4 \mapsto a_1][s_3 \mapsto a_2] & 0 \\ [s_4 \mapsto a_2][s_3 \mapsto a_2] & 2 \end{bmatrix}$$

**Iteration #5.** Update $s_2, a_2$.

$$Q[s_2 a_2] = R(s_2, a_2) + \gamma[p(s_3|s_2, a_2)\text{ConQ}[s_3]]$$
$$= \text{ConQ}[s_3]$$

$$= \left[\begin{array}{c|c}
\text{Partition of } \Theta & \text{Q-values} \\
\hline
[s_3 \mapsto a_1] & 0 \\
[s_4 \mapsto a_1][s_3 \mapsto a_2] & 0 \\
[s_4 \mapsto a_2][s_3 \mapsto a_2] & 2
\end{array}\right]$$

$$\text{ConQ}[s_2 a_2] = \left[\begin{array}{c|c}
\text{Partition of } [s_2 \mapsto a_2] & \text{Q-values} \\
\hline
[s_3 \mapsto a_1][s_2 \mapsto a_2] & 0 \\
[s_4 \mapsto a_1][s_3 \mapsto a_2] \cap [s_2 \mapsto a_2] = \emptyset & - \\
[s_4 \mapsto a_2][s_3 \mapsto a_2] \cap [s_2 \mapsto a_2] = \emptyset & -
\end{array}\right]$$

$$= \left[\begin{array}{c|c}
\text{Partition of } [s_2 \mapsto a_2] & \text{Q-values} \\
\hline
[s_3 \mapsto a_1][s_2 \mapsto a_2] & 0
\end{array}\right]$$

The witness oracle checks feasibility of $[s_3 \mapsto a_2][s_2 \mapsto a_2]$ by solving a system of linear inequalities. In this case,

$$[s_2 \mapsto a_2] \implies \theta \cdot \phi(s_2, a_2) > \theta \cdot \phi(s_2, a_1) \implies \theta_1 > 0,$$
$$[s_3 \mapsto a_2] \implies \theta \cdot \phi(s_3, a_2) > \theta \cdot \phi(s_3, a_1) \implies \theta_1 < 0.$$

Hence the assignment of these two policy actions to these two states is infeasible and PCVI eliminates those two entries in the $\text{ConQ}[s_2 a_2]$ table.

$$\text{ConQ}[s_2] = \left[\begin{array}{c|c}
\text{Partition of } \Theta & \text{Q-values} \\
\hline
[s_2 \mapsto a_1] & 0 \\
[s_3 \mapsto a_1][s_2 \mapsto a_2] & 0
\end{array}\right]$$

**Iteration #6.** Update $s_2, a_1$.

$$Q[s_2 a_1] = R(s_2, a_1) + \gamma[p(\perp |s_2, a_1)\text{ConQ}[\perp]]$$
$$= \text{ConQ}[\perp]$$

$$= \left[\begin{array}{c|c}
\text{Partition of } \Theta & \text{Q-values} \\
\hline
\Theta & 0
\end{array}\right]$$

$$\text{ConQ}[s_2 a_1] = \left[\begin{array}{c|c}
\text{Partition of } [s_2 \mapsto a_1] & \text{Q-values} \\
\hline
\Theta \cap [s_2 \mapsto a_1] = [s_2 \mapsto a_1] & 0
\end{array}\right]$$

$$\text{ConQ}[s_2] = \left[\begin{array}{c|c}
\text{Partition of } \Theta & \text{Q-values} \\
\hline
[s_2 \mapsto a_1] & 0 \\
[s_3 \mapsto a_1][s_2 \mapsto a_2] & 0
\end{array}\right]$$

**Iteration #7.** Update $s_1, a_2$.

$$Q[s_1 a_2] = R(s_1, a_2) + \gamma[p(s_2|s_1, a_2)\text{ConQ}[s_2]]$$
$$= \text{ConQ}[s_2]$$

$$= \left[\begin{array}{c|c}
\text{Partition of } \Theta & \text{Q-values} \\
\hline
[s_2 \mapsto a_1] & 0 \\
[s_3 \mapsto a_1][s_2 \mapsto a_2] & 0
\end{array}\right]$$

$$\text{ConQ}[s_1 a_2] = \left[\begin{array}{c|c}
\text{Partition of } [s_1 \mapsto a_2] & \text{Q-values} \\
\hline
[s_2 \mapsto a_1][s_1 \mapsto a_2] & 0 \\
[s_3 \mapsto a_1][s_2 \mapsto a_2][s_1 \mapsto a_2] & 0
\end{array}\right]$$

$$\text{ConQ}[s_1] = \left[\begin{array}{c|c}
\text{Partition of } \Theta & \text{Q-values} \\
\hline
[s_1 \mapsto a_1] & 0 \\
[s_2 \mapsto a_1][s_1 \mapsto a_2] & 0 \\
[s_3 \mapsto a_1][s_2 \mapsto a_2][s_1 \mapsto a_2] & 0
\end{array}\right]$$

**Iteration #8.** Update $s_1, a_1$.

$$\mathtt{Q}[s_1 a_1] = R(s_1, a_1) + \gamma[p(\bot \,|s_1, a_1)\mathtt{ConQ}[\bot] \oplus p(s_4|s_1, a_1)\mathtt{ConQ}[s_4]]$$
$$= 0.3 + 0.9\mathtt{ConQ}[\bot] \oplus 0.1\mathtt{ConQ}[s_4]$$

$$= \begin{bmatrix} \begin{array}{c|c} \text{Partition of } \Theta & \text{Q-values} \\ \hline \Theta \cap [s_4 \mapsto a_1] & 0.3 + 0.9\mathtt{ConQ}[\bot](\Theta) + 0.1\mathtt{ConQ}[s_4]([s_4 \mapsto a_1]) = 0.3 \\ \Theta \cap [s_4 \mapsto a_2] & 0.3 + 0.9\mathtt{ConQ}[\bot](\Theta) + 0.1\mathtt{ConQ}[s_4]([s_4 \mapsto a_2]) = 0.5 \end{array} \end{bmatrix}$$

$$\mathtt{ConQ}[s_1 a_1] = \begin{bmatrix} \begin{array}{c|c} \text{Partition of } [s_1 \mapsto a_1] & \text{Q-values} \\ \hline [s_4 \mapsto a_1][s_1 \mapsto a_1] & 0.3 \\ {[s_4 \mapsto a_2][s_1 \mapsto a_1]} & 0.5 \end{array} \end{bmatrix}$$

$$\mathtt{ConQ}[s_1] = \begin{bmatrix} \begin{array}{c|c} \text{Partition of } \Theta & \text{Q-values} \\ \hline [s_2 \mapsto a_1][s_1 \mapsto a_2] & 0 \\ {[s_3 \mapsto a_1][s_2 \mapsto a_2][s_1 \mapsto a_2]} & 0 \\ {[s_4 \mapsto a_1][s_1 \mapsto a_1]} & 0.3 \\ {[s_4 \mapsto a_2][s_1 \mapsto a_1]} & 0.5 \end{array} \end{bmatrix}$$

In iteration #9, we would update $s_4, a_2$ but none of the $\mathtt{Q}$ or $\mathtt{ConQ}$ data structures change. Likewise, subsequent iteration updates to any other $s, a$ does not change these tables. Thus we have convergence and can recover the optimal admissible policy as follows. For initial state $s_1$ we look up table $\mathtt{ConQ}[s_1]$ and find that the feasible set with highest value is $X^* = [s_4 \mapsto a_2][s_1 \mapsto a_1]$ with value 0.5. A feasible parameter $\theta^* = (-2, 0.5)$ can also be found via, for example, solving a system of linear inequalities by linear programming. Fig. 8 summarizes the backups performed by PCVL and the resulting $\mathtt{ConQ}$ tables.

## A.8 PCQL with $\mathtt{ConQ}$ regression

While Policy-class Q-Learning allows one to generalize in the sense that a parameterized policy is returned, it does not explicitly model a corresponding Q-function. Such a Q-function allows one to predict and generalize state-action values of unobserved states. For example, when an initial state is never observed in the training data, the Q-function can be used to assess how good a particular cell or equivalence class of policies are.

This is the general idea behind our heuristic. It keeps a global collection of information sets that are gradually refined based on training data. The information set contains both the constraints defining a cell and a regressor that predicts *consistent* Q-values for that cell's policies. When there are too many information sets, one can use a pruning heuristic, such as removing information sets with low Q-values. The backup operation must respect the consistency requirement: a cell's Q-regressor gets updated only if $[s \mapsto a]$ is consistent with its constraints. As discussed earlier, pruning feasible information sets may result in no updates to any cell given a sample transition (since no cell may be consistent). But this can be avoided if our heuristic merges instead of deleting cells.

The algorithm pseudo-code is given in Algorithm 3. Every information set is a pair $(X, w)$ where $X$ is a set of parameters consistent with some set of constraints and $w$ are the weights of a Q-regressor. In general $w$ may be from a different function class than $\Theta$. The Q-labels that $w$ learns from is generated in the consistent manner stated above, that is, it predicts the consistent Q-value $\mathtt{ConQ}_w(s, a)$ for any state-action pair. While it produces values for any $(s, a)$ that is inconsistent with $X$, such values are not used for backups or finding an optimal policy.

## A.9 Constructing consistent labels

We formulate the problem of consistent labeling as an mixed integer program (MIP).

Assume a batch of training examples $B = \{(s_t, a_t, r_t, s'_t)\}_{t=1}^T$, and a current regressor $\widetilde{Q}$ used to create "bootstrapped" labels. The nominal Q-update generates labels of the following form for each pair $(s_t, a_t)$: $q_t = r_t + \gamma \max_{a'_t} \widetilde{Q}(s'_t, a'_t)$. The updated regressor is trained in supervised fashion with inputs $(s_t, a_t)$ and label $q_t$.

To ensure policy class consistency, we must restrict the selection of the maximizing action $a'_t$ so that:

**Algorithm 3** Policy-Class Q-Learning with Regression

---

**Input:** Batch $B = \{(s_t, a_t, r_t, s'_t)\}_{t=1}^T$, $\gamma$, $\Theta$, scalars $\alpha_t^{sa}$, initial state $s_0$.
1: Initialize information sets $I \leftarrow \{(\Theta, w_\Theta)\}$.
2: $visited \leftarrow \emptyset$
3: **for** $(s, a, r, s') \in B$, $t$ is iteration counter **do**
4:     If $s \notin visited$ then Refine($s$)
5:     If $s' \notin visited$ then Refine($s'$)
6:     **for** $(X, w) \in I$ **do**
7:       **if** $X \cap [s \mapsto a] \neq \emptyset$ **then**
8:         $w \leftarrow w + \alpha_t^{sa}(r + \gamma \texttt{ConQ}_w(s', \pi_X(s')) - \texttt{ConQ}_w(s, a))\nabla_w \texttt{ConQ}_w(s, a)$
9:       **end if**
10:      Prune $I$ if too many information sets
11:     **end for**
12: **end for**
13:
14: /* Then recover an optimal policy */
15: If $s_0 \notin visited$ then Refine($s_0$)
16: $(X^*, w^*) \leftarrow \text{argmax}_{(X, w)} \texttt{ConQ}_w(s_0, \pi_X(s_0))$
17: Select some witness $\theta^* \in X^*$ then return $\pi_{\theta^*}$
18:
19: **Procedure** Refine($s$)
20:     $I_{new} \leftarrow \emptyset$
21:     **for** $(X, w) \leftarrow I.\text{pop()}$ **do**
22:       $X_i \leftarrow X \cap [s \mapsto a_i]$ for all $a_i$
23:       If $X_i \neq \emptyset$ then $I_{new}.\text{add}((X_i, w))$
24:     **end for**
25:     $I \leftarrow I_{new}$
26:     $visited.\text{add}(s)$

---

- $\cap_t [s'_t \mapsto a'_t] \neq \emptyset$ (i.e., selected maximizing actions are mutually consistent); and

- $[s_t \mapsto a_t] \cap [s'_t \mapsto a'_t] \neq \emptyset$, for all $t$ (i.e., choice at $s'_t$ is consistent with taking $a_t$ at $s_t$).

We construct a consistent labeling by finding an assignment $\sigma : s'_t \rightarrow A(s'_t)$, assuming some reasonable maximization objective that satisfies these constraints. We illustrate this using the sum of the resulting labels as the optimization objective, though other objectives are certainly possible.

With a linear approximator, the problem can be formulated as a (linear) mixed integer program (MIP). For any parameter vector $\theta \in \Theta$, we write $\theta(s, a)$ to denote the linear expression of $Q(s, a; \theta)$. To meet the first requirement, we assume a single (global) parameter vector $\theta_g$. For the second, we have a separate parameter vector $\theta_t$ for each training example $(s_t, a_t, r_t, s'_t)$. We have variables $q_t$ representing the bootstrapped (partial) label for each training example. Finally, we have an indicator variable $\mathbb{I}_{a'}^t$ for each $t \leq T$, $a' \in A(s')$: this denotes the selection of $a'$ as the maximizing action for $s'_t$. We assume rewards are non-negative for ease of exposition. The following IP

over the interval becomes linear:

$$\max_{\mathbb{I}_{a'}^t, q_t, \theta_g, \theta_t} \sum_{t \leq T} q_t \tag{11}$$

$$s.t. \quad \theta_t(s_t, a_t) \geq \theta_t(s_t, b) \forall b \in A(s_t), \forall t \tag{12}$$

$$\theta_t(s'_t, a'_t) \geq \mathbb{I}_{a'}^t \theta_t(s'_t, b') \forall b' \in A(s'_t), \forall t \tag{13}$$

$$\theta_g(s'_t, a'_t) \geq \mathbb{I}_{a'}^t \theta_g(s'_t, b') \forall b' \in A(s'_t), \forall t \tag{14}$$

$$q_t = r_t + \gamma \sum_{a' \in A(s'_t)} \mathbb{I}_{a'}^t \theta_g(s'_t, a'_t) \forall t \tag{15}$$

$$\sum_{a' \in A(s'_t)} \mathbb{I}_{a'}^t = 1, \forall t \tag{16}$$

The nonlinear term $\mathbb{I}^t_{a'}\theta_g(s'_t, a'_t)$ in the fourth constraint can be linearized trivially using a standard transformation since it is the product of a real-valued and a binary (0-1) indicator variable. This IP can be tackled heuristically using various greedy heuristics as well.

## A.10 Discounting Hyperparameter Experiments

Our experiments use the Atari learning environment [3] to demonstrate the impact of changing the training discount factor, $\gamma_\ell$, used in Q-learning on the resulting greedy policy. The delusional bias is particularly pronounced as $\gamma_\ell$ becomes close to 1, since incorrect temporal difference terms become magnified in the updates to the function approximator. As shown above, the delusional bias problem cannot be solved solely by choosing the right $\gamma_\ell$; instead we show here that sweeping a range of $\gamma_\ell$ values can reveal the discounting paradoxes in realistic environments, while also showing how $\gamma_\ell$ tuning might mitigate some of the ill effects. Recall that it is possible for a larger $\gamma_\ell$ might lead to a better policy when evaluated on a smaller $\gamma_e$—this is in contrast to [15, 16] which suggest using a smaller $\gamma_\ell \leq \gamma_e$. In fact we see both phenomena.

We use a state-of-the-art implementation of Deep Q-Networks (DQN) [19], where we trained $Q_{\hat{\theta}\gamma_\ell}$ by varying $\gamma_\ell$ used in the Q-updates while holding other hyperparameters constant. We use $\varepsilon$Greedy with $\varepsilon = 0.07$ as the behaviour policy, and run training iterations until the training scores converge. Each iteration consists of at most $2.5 \times 10^5$ training steps (i.e., transitions), with max steps per episode set to $27 \times 10^3$. We use experience replay with a buffer size of $10^6$ and a mini-batch size of 32. We evaluate converged Q-functions using the corresponding greedy policy[5] over episodes of $1.2 \times 10^6$ steps. We do 5 training restarts for each game tested.

We vary $\gamma_\ell$ to reflect varying effective horizon lengths, $h = 1/(1 - \gamma_\ell)$, of $h \in \{100 - 10n : n = 0, 1, \dots, 8\}$. ($\gamma_\ell > 0.99$ tends to cause divergence. For evaluation, we use the same values for $\gamma_e$, plus $\gamma_e \in \{0.995, 1\}$—the latter is commonly reported as total undiscounted return.

Figs. 9 and 10 show (normalized) returns across four benchmark Atari games, averaged across the 5 training restarts. The implementation we use is known to have low variability in performance/value in Atari once DQN has converged. Indeed, training scores converged to very similar values across restarts (i.e., the return of $\varepsilon$Greedy($\varepsilon = 0.07$) with $\gamma_e = 1$); however, the resulting greedy policies can still exhibit differences (e.g., SpaceInvaders, $\gamma_\ell = 0.9833$ vs. $\gamma_\ell = 0.9857$ rows).

These heatmaps reveal several insights. Training with smaller $\gamma_\ell$ usually results in better policies, even when evaluated at maximum horizons, e.g., if we compare the first two rows to the last two across all four games. This is particularly true in Seaquest, where the two smallest $\gamma_\ell$ values give the best policies across a range of $\gamma_e$. The last row of Seaquest shows the most non-myopic policy performs worst.

Most critically, the heatmaps are clearly not diagonally dominant. One might expect each column to be single-peaked at the "correct" discount factor $\gamma_\ell = \gamma_e$, which should exhibit the highest (normalized) discounted return, with returns are monotonically non-decreasing when $\gamma_\ell < \gamma_e$ and monotonically non-increasing when $\gamma_\ell > \gamma_e$. But this does not occur in any domain. Qbert is perhaps the closest to being diagonally dominant, where smaller (resp. larger) $\gamma_\ell$ tend to perform better for smaller (resp. larger) $\gamma_e$.

In SpaceInvaders, a myopic policy $\pi_1$ (trained with $\gamma_\ell = 0.9667$) is generally better, than a more non-myopic policy $\pi_2$ (e.g. $\gamma_\ell = 0.9889$). Policy $\pi_1$ consistently achieves the best average evaluation score for for a variety of $\gamma_e$. about 29.7% better than $\pi_2$ for $\gamma_e = 1$. Figs. 11 and 12 show additional heatmaps demonstrating a different view of relative performance. Our results show that the "opposite" counter-example (Sec. 3.2) can also arise in practice. See Fig. 11, Qbert heatmap—in particular policy trained with $\gamma_\ell = 0.975$ generally performs better than policy trained with $\gamma_\ell = 0.99$. Prior work has only uncovered outperformance of myopically trained policies (though not with Q-learning), while we show that the opposite can also occur.

Figs. 11 and 12 have been normalized across each column so that average discounted sum of rewards in each column are in the unit interval. This allows for easier comparison across the different policies learned from varying $\gamma_\ell$. Note that this also exacerbates potentially small differences in the values (e.g. when original values in columns lie in a small range).

In Seaquest, one can clearly see the relative outperformance of models trained with $\gamma_\ell = 0.95, 0.9667$ over more non-myopic models (e.g. last three rows). For Qbert, the best policy for $\gamma_e < 0.995$ is the policy trained with $\gamma_\ell = 0.975$ while the best policy for $\gamma_e = 1$ is trained with $\gamma_\ell = 0.9857$. For Pong, at first place, the performance of model trained with $\gamma_\ell = 0.95$ gradually becomes worse. But this in fact is due to scaling by ever larger discount factors—if the undiscounted scores are negative then larger discount scaling only decreases its performance.

Figure 9: Each entry $s_{ij}$ of left (red) heatmap shows the normalized scores for evaluating $\varepsilon\mathsf{Greedy}(Q_{\hat{\theta}^{\gamma_\ell}}, \varepsilon = 0.005)$ (row $i$) using $\gamma_e$ (column $j$). The normalization is across each column: $s_{ij} \leftarrow (1 - \gamma_e)s_{ij}$. The right heatmaps (blue) show avg. unnormalized undiscounted total returns. Results averaged over 5 training restarts.

Figure 10: Similar to Fig. 9 with heatmaps for SpaceInvaders and Seaquest.

Figure 11: Each entry $s_{ij}$ of a heatmap shows the scores normalized to the unit interval for evaluating $\varepsilon\text{Greedy}(Q_{\hat{\theta}^{\gamma_\ell}}, \varepsilon = 0.005)$ (row $i$) using $\gamma_e$ (column $j$). The normalization is across each column: $s_{ij} \leftarrow (s_{ij} - \min_k s_{kj})/(\max_k s_{kj} - \min_k s_{kj})$. Results averaged over 5 training restarts.

Figure 12: Similar to Fig. 11 with heatmaps for SpaceInvaders and Seaquest.

## Footnotes

[3]Note that we use these discount rates of 0 and 1 to illustrate how extreme the paradox can be—there is nothing intrinsic to the paradox that depends on the use of the purely myopic variant versus the infinite-horizon variant.

[4] If $\hat{\theta}_2 < 0$, the greedy policy is $\pi_{a_1}$, which induces the stationary distribution $\mu_{a_1}(s_1, a_1) = 1 - \varepsilon$; $\mu_{a_1}(s_1, a_2) = \varepsilon$; $\mu_{a_1}(s_2, a_1) = (1-\varepsilon)^2$; $\mu_{a_1}(s_2, a_2) = \varepsilon(1-\varepsilon)$; $\mu_{a_1}(s_2', a_1) = \varepsilon(1-\varepsilon)$; and $\mu_{a_1}(s_2', a_2) = \varepsilon^2$. When $\hat{\theta}_2 > 0$, the greedy policy $\pi_{a_2}$ induces a similar distribution, but with $\varepsilon$ and $1 - \varepsilon$ switched.

[5] We use $\varepsilon$Greedy with $\varepsilon = 0.005$ to prevent in-game loops.