[Reviews · NeurIPS 2018]

Reviewer 1



The paper defines a new type of reinforcement learning algorithm, which takes account of the imperfections of the function approximator and tries to obtain the best policy available given these imperfections rather than assuming no imperfections exist, thus avoiding pathologies arising when we assume a flawed approximate is perfect. The quality of this paper is really good. It introduces a new type of RL algorithm, which is clearly motivated and solid. The weaker points are: 1. The complexity of the defined algorithm seems too high for it to be immediately applicable to interesting problems. However, the authors honestly acknowledge this and provide a bound saying that, even though costly, the algorithm is still polynomial as opposed to exponential, as might have initially been expected given the doubling property of the equation below line 191. 2. The authors do not study the kind of task where I think their ideas would have the most use: tasks where we have a lot of computation when the policy is being prepared but very little when it is executed (for example because it has to be embedded in a cheap hardware item). The proposed algorithm is super-well suited for this because the learning is expensive put the obtained policy is very small / has a low degree of complexity. The potential significance of the paper is large (although significant obstacles remain to make this practical). It is definitely a good match for NIPS. The clarity of the language and the exposition of the math is very good. On the balance, I feel that this paper would clearly benefit the NIPS community despite the mentioned shortcomings. At the end of the day, the paper introduces genuinely novel concepts and I strongly believe that it would be unfair to demand an empirical evaluation at the same scale as papers which make relatively minor adjustments to existing techniques. I hope more follow-up work is published exploring this type of ideas in a more practical setting, but the current paper is already so packed that it would be disproportionate to expect such work to be included in this submission. Moreover, I hope that he authors will also submit this work (or an extension) to a journal and possibly spend some more time outlining the intuitions, possibly with a visual walk-through showing how the proposed algorithm partitions the parameter space for some toy problem.

Reviewer 2



UPDATE: Thank you to the authors for a well thought out response. I definitely encourage you to compile the list of counterexamples that are or are not the result of delusion. I think it will be a very important list for the theory community. Also, thanks for the discounting example, that helped. Good point on the Boyan example and ADP divergence and the expanded discussion. Summary: The paper defines the concept of delusional bias’’, the fact that the Bellman operator (or the sample-based version in Q-learning) combined with a function approximator, can backup values from a policy that exist in the realizable space of the approximator, essentially assuming that two actions that it cannot simultaneously take can be chosen at the same time. The authors propose a new operator that uses a partitioning of the parameter space into information sets. The operator essentially maintains a large set of potential combinations of the partitions and q values for each of the sets, eventually choosing the maximum after convergence. This requires significant (but sometimes polynomially bounded) computation, although in many cases an intractable witness procedure is needed for complex function approximators. Review: This is a potentially ground-breaking paper. I have comments below on ways to improve the presentation of the paper and some minor mathematical notes but overall, I think the authors did an excellent job on a long standing foundational problem in RL. The paper cites some important foundational work in the introduction but does not explicitly state which existing negative function approximation results are explained by delusional bias. Is Baird’s counterexample a result of delusional bias and therefore solved by the new algorithm? It would be better to explicitly mention which of the classical results can (and cannot) be explained by this phenomenon. I do not see the direct link between delusional bias and the “discounting paradox” described in the paper. The example given mixes a discounted MDP with an undiscounted one, and really those two problems are fundamentally different. Is there an example where both discounts are < 1? If so, is the conjecture of the paper that small discount factors essentially truncate the backups and therefore somehow enforce a partitioning? I don’t understand why that would be the case. Clearly changing the discount factor changes the solution, but it is not clear why the paradox is really related to delusional bias. I thought perhaps the most important line in the paper was on line 235 where the authors note that their result is actually consistent with Petrik’s finding on the NP-Hardness of finding the best function approximator. The authors essentially point out that Petrik was considering all possible action choices, and therefore maybe asking the wrong question. Perhaps minimizing Bellman error isn’t all that important if one is considering actions that can’t actually be taken. Perhaps that can be spelled out better here. To make room for some of the explanations above, I suggest cutting down much of the speculation in the section 5. These are interesting conjectures, but do not have any real empirical evidence behind them. If a particular example of one of these three approaches could be demonstrated here it would certainly improve the paper. In section 3.2, it would be good to point out explicitly that there are different consequences of delusion in q-learning versus ADP. Specifically, q-learning can actually diverge while I think the claim here is that ADP converges, but to something non-optimal given the approximator constraints, right? That point should be made explicit. Line 182 – the parentheses around the full sentence can be dropped. It’s a full idea / sentence.

Reviewer 3



This paper considers the limitations of using a restricted policy representation in RL. In some cases, the class of approximation functions used to represent the policy cannot jointly realize an optimal temporal sequence of actions. The authors introduce this so-called "delusional bias" phenomenon, characterize it using extensive examples and derive a possible algorithm to overcome its effects. The algorithm is shown to converge to an optimal policy efficiently in the tabular case and several heuristics are proposed to scale up it to more complex scenarios. Globally, the paper is well written. The main strength is the characterization of this "delusional bias" phenomenon and the extensive analysis, which is original and very interesting. The weaknesses of the paper are (i) the presentation of the proposed methodology to overcome that effect and (ii) the limitations of the proposed methods for large-scale problems, which is precisely when function approximation is required the most. While the intuition behind the two proposed algorithms is clear (to keep track of partitions of the parameter space that are consistent in successive applications of the Bellman operator), I think the authors could have formulated their idea in a more clear way, for example, using tools from Constraint Satisfaction Problems (CSPs) literature. I have the following concerns regarding both algorithms: - the authors leverage the complexity of checking on the Witness oracle, which is "polynomial time" in the tabular case. This feels like not addressing the problem in a direct way. - the required implicit call to the Witness oracle is confusing. - what happens if the policy class is not realizable? I guess the algorithm converges to an \empty partition, but that is not the optimal policy. minor: line 100 : "a2 always moves from s1 to s4 deterministically" is not true line 333 : "A number of important direction" -> "A number of important directions" line 215 : "implict" -> "implicit" - It is hard to understand the figure where all methods are compared. I suggest to move the figure to the appendix and keep a figure with less curves. - I suggest to change the name of partition function to partition value. [I am satisfied with the rebuttal and I have increased my score after the discussion]